# On the Effects of Data Scale on UI Control Agents

Wei Li[1], William Bishop[1], Alice Li[1], Chris Rawles[1], Folawiyo Campbell-Ajala[1],
Divya Tyamagundlu[2], and Oriana Riva[1]

[1]*Google DeepMind*
[2]*Google*

## Abstract

Autonomous agents that control user interfaces to accomplish human tasks are emerging. Leveraging LLMs to power such agents has been of special interest, but unless fine-tuned on human-collected task demonstrations, performance is still relatively low. In this work we study whether fine-tuning alone is a viable approach for building real-world UI control agents. To this end we collect and release a new dataset, ANDROIDCONTROL, consisting of 15,283 demonstrations of everyday tasks with Android apps. Compared to existing datasets, each ANDROIDCONTROL task instance includes both high and low-level human-generated instructions, allowing us to explore the level of task complexity an agent can handle. Moreover, ANDROIDCONTROL is the most diverse UI control dataset to date, including 14,548 unique tasks over 833 Android apps, thus allowing us to conduct in-depth analysis of the model performance in and out of the domain of the training data. Using the dataset, we find that when tested in domain fine-tuned models outperform zero and few-shot baselines and scale in such a way that robust performance might feasibly be obtained simply by collecting more data. Out of domain, performance scales significantly more slowly and suggests that in particular for high-level tasks, fine-tuning on more data alone may be insufficient for achieving robust out-of-domain performance.

## 1 Introduction

Recent work has studied how large language models (LLMs) can be leveraged to build UI control agents [18, 43, 39, 17, 7] that accomplish human tasks by interacting with a digital device environment. These agents perceive the state of the device by observing its screen (from screenshots or application UI trees), and generate actions (click, type, scroll, etc.) that are executed through the device's user interface. Tasks, specified in natural language, can range from configuring device settings and sending emails, to navigating shopping websites and planning a trip.

While progress is rapidly advancing, absolute performance of UI control agents that leverage pre-trained LLMs without fine-tuning on task demonstrations is still relatively low. When tested in real-world environments, where agents control everyday applications and websites, recently-reported task success rates range from 12% on desktop applications [38] to 46% on mobile applications [3]. In contrast, agents that leverage models fine-tuned for task execution [25, 10], achieve even 80% [10] success rate, when tested on websites and tasks similar to what they are trained on.

While the pattern of collecting new datasets and fine-tuning shows promise, there are at least two important unanswered questions. First, to the best of our knowledge no prior work has examined the question of scaling: how much data must be collected in order to obtain a given performance level with fine-tuned models. This question is particularly important because human demonstrations

38th Conference on Neural Information Processing Systems (NeurIPS 2024) Track on Datasets and Benchmarks.

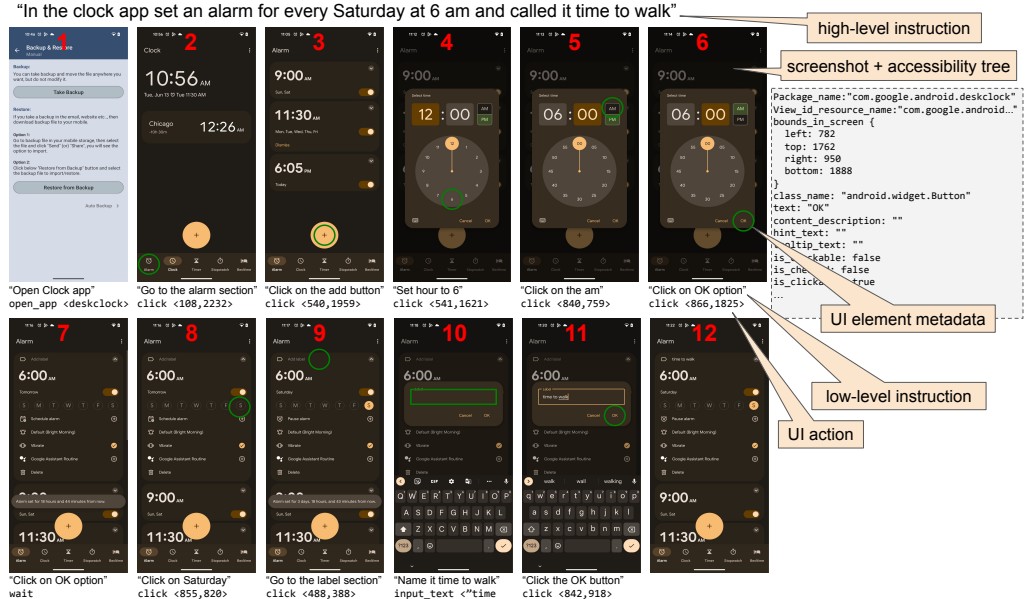

Figure 1: An example task demonstration contained in ANDROIDCONTROL. Green circles/rectangles highlight the action on the screen. Red numbers are added only for illustration purposes.

of UI interactions for fine-tuning are time consuming and expensive to collect. Understanding how performance scales, both in domain and out of the domain of the collected demonstrations (unseen tasks and unseen applications), is important for determining whether fine-tuning alone is a viable path towards deploying UI control agents in the real world. Therefore, one of the main goals of this work is to rigorously quantify how performance of fine-tuned agents scales, both in and out of domain, as the amount of data used for fine-tuning is increased.

Second, it is not clear the level of task complexity fine-tuning might be fruitfully applied to. Conceptually, UI control agents must both decompose a high-level goal into a set of small atomic actions and execute ("ground") those actions in a device screen. While the high-level reasoning with LLMs, required for determining how to accomplish high-level goals, is still an open problem in artificial intelligence [35, 34, 42, 45, 19, 41, 36], the set of low-level actions (clicking, typing, etc...) required to execute tasks are more constrained, and general agents capable of robust grounding across domains might be approachable via fine-tuning. Therefore, a second goal of this work is to quantify the scaling of fine-tuning for agents performing both high-level and low-level tasks.

Rigorously quantifying scaling in these ways requires a carefully constructed dataset. To this end, we introduce ANDROIDCONTROL, a large-scale dataset of 15,283 demonstrations of tasks performed by humans in Android apps. Figure 1 shows an example data sample. Compared to existing datasets [7, 27], for every task ANDROIDCONTROL provides both the high- and low-level human-generated instructions describing it. This is essential to investigate the level of task complexity a model can handle and also provides richer supervision during training. ANDROIDCONTROL is also the most diverse UI control dataset that exists today, including 14,548 unique tasks over 833 different Android apps, thus allowing us to generate multiple test splits for measuring performance in and out of domain. As a resource to the community, we make ANDROIDCONTROL publicly available.[1]

Overall, we make the following contributions: *(i)* we collect and release ANDROIDCONTROL, a new UI control dataset whose size, structure and diversity advances previous datasets, *(ii)* we use ANDROIDCONTROL to quantify how fine-tuning with demonstrations scales when applied to both low- and high-level tasks and to tasks in and out of the domain of the training data, and *(iii)* we compare fine-tuning to various zero-shot and few-shot baselines, finding that fine-tuning scales favorably in domain, but out of domain, it requires one or two orders of magnitude more data to obtain robust performance on high-level tasks, suggesting that additional approaches may be beneficial for obtaining agents which robustly perform out-of-domain high-level tasks.

---

[1] https://github.com/google-research/google-research/tree/master/android_control

| Dataset | Platform | # Human demos | # Unique instr. | # Apps or websites | # Task steps | UI tree? | Screen? | High-level instr. | Low-level instr. |
|---|---|---|---|---|---|---|---|---|---|
| MiniWoB++ [31] | Web (synthetic) | 17,971 | 100 | 114 | 2.3 | ✓ | ✗ | ✗ | ✓ |
| WebShop [40] | Web | 1,566 | 1,566 | 1 | 11.3 | ✗ | ✓ | ✓ | ✗ |
| UIBert [2] | Android | 16,660 | - | - | 1.0 | ✓ | ✓ | ✗ | ✓ |
| PixelHelp [22] | Android | 187 | 187 | 4 | 4.2 | ✓ | ✗ | ✓ | ✓ |
| UGIF [33] | Android | 523 | 523 | 12 | 5.3 | ✓ | ✓ | ✓ | ✓ |
| MoTIF [6] | Android | 4,707 | 276 | 125 | 4.5 | ✓ | ✓ | ✓ | ✓ |
| Mind2Web [7] | Web | 2,350 | 2,350 | 137 | 7.3 | ✓ | ✓ | ✓ | ✗ |
| AitW [27] | Android | 715,142 | 30,378 | 357 | 6.5 | ✗ | ✓ | ✓ | ✗ |
| WebVoyager [12] | Web | 643 | 643 | 15 | - | ✓ | ✓ | ✓ | ✗ |
| WebLINX [24] | Web | 2,337 | 2,377 | 155 | 43.0 | ✓ | ✓ | ✓ | ✗ |
| **ANDROIDCONTROL** | Android | 15,283 | 14,548 | 833 | 5.5 | ✓ | ✓ | ✓ | ✓ |

Table 1: Comparison of ANDROIDCONTROL to existing UI control datasets. We consider device platform, size (as number of task demonstrations), diversity (as number of unique task instructions and apps/websites), average number of steps in a task, how the UI state is captured (UI tree vs. screenshot), and whether tasks are described through high-level instructions or sequences of low-level commands.

## 2  Related work

**UI control datasets**    Table 1 compares ANDROIDCONTROL to existing UI control datasets. The structure of these datasets is similar. They consist of a natural language task description and a human-recorded demonstration, in the form of a sequence of UI actions (click, type, swipe, etc.) and associated UI states. What differentiates these datasets is mainly whether they are single-step (as in grounding referring expressions datasets like UIBert [2]), whether the task description is expressed as a high-level goal or as a sequence of low-level step instructions, and how the UI state is represented (screenshot vs. UI tree). Three features make ANDROIDCONTROL unique. First, for every task, it contains both low-level and high-level instructions generated by human annotators. While a few other datasets contain both these types of annotation, their low-level instructions are either synthetically generated (as in MoTIF [6]) or are limited to one action type (only click actions [22, 33]). In addition to bringing richer language supervision during training, the availability of human-generated low-level instructions allows us to test UI control agents on different levels of task complexity. Second, if we consider the number of unique task instructions and the number of human demonstrations, ANDROIDCONTROL is the second-largest UI control dataset to date, second only to AitW [27].[2] The diversity of task scenarios present in ANDROIDCONTROL is its third differentiating feature: ANDROIDCONTROL includes tasks from 833 different Android apps, 6 times more than popular datasets like Mind2Web [7] and almost 5 times more than AitW. This diversity makes ANDROIDCONTROL optimal for realistic, out-of-domain analysis. Note that Mind2Web also provides out-of-domain splits but given its smaller size (2,350 tasks over 137 websites, with a train split of 1k demonstrations) is not suitable for a scaling analysis.

In addition to the datasets listed in Table 1, recent work proposes interactive testing environments for UI control agents [31, 46, 18, 38, 28, 4] where the environment provides the agents with reward signals. These environments are designed for online testing and are limited to no more than 20 applications or websites. The only exception is MiniWob [31] for which task demonstrations have been collected, but the environment consists of much simplified, synthetic websites.

**UI control agents**    Early UI control agents were trained from scratch using behavioural cloning [15, 21, 22] or reinforcement learning [23, 11]. Current UI agents use pre-trained LLMs and multimodal models. One line of work prompts LLMs in a zero-shot or few-shot regime [39, 27, 12, 18, 17, 44]. Another line of work relies on fine-tuning which is applied end to end [25] or to build specific model capabilities, such as identifying the interactable UI elements in a webpage [43, 10, 7]. To name a few, SeeAct [43], which we use in our evaluation, is a web agent that leverages large multimodal models to understand text and visual elements on webpages. The best-performing SeeAct agent relies on a fine-tuned cross-encoder model to select candidates web elements for interaction. WebGPT [25]

---

[2]AitW is heavily web focused. If we consider the AitW episodes involving Android apps, we obtain 505k episodes across 167 apps, for a total of 1.6k unique task instructions, a much smaller number of ANDROIDCONTROL. AitW also does not contain application UI trees, thus making the UI state representations incomplete.

fine-tunes GPT-3 to learn to use a web browser. WebAgent [10] pre-trains a T5 model to extract HTML snippets and leverages Flan-U-PaLM to generate Python code to control a web environment. Synapse [44] introduces a trajectory-as-exemplar prompting method where memory of previous interactions allows the agent to perform complex, multi-step tasks.

**Domain generalization**    As evidenced by various LLM studies [5, 16, 13], scaling model and data size for the training leads to steady improvements in domain generalization. On the other hand, when transferring a pre-trained model to a downstream task through fine-tuning, while in-distribution performance improves, a reduction in the robustness to distribution shifts is observed [20, 37, 1]. In this work, we empirically study how scaling data size in fine-tuning affects in-domain and out-of-domain performance of UI control agents. While prior work has tested UI agents using out-of-domain test splits [7, 6], to the best of our knowledge a data scale analysis has not been conducted. To minimize training cost and to maintain the out-of-domain generalization, we evaluate also the option of not fine-tuning, by evaluating multiple zero-shot and few-shot baselines.

## 3    The ANDROIDCONTROL dataset

The collection of the ANDROIDCONTROL dataset is motivated by our dual goal of studying (i) how scaling data size for fine-tuning UI control models affects in-domain and out-of-domain performance, and (ii) the level of task complexity these fine-tuned models can be effective for.

### 3.1    Data collection

We collect ANDROIDCONTROL using crowdsourcing over the course of a year. The data collection starts by giving crowdworkers generic feature descriptions for apps from 40 different categories (Figure 2). These descriptions are generated using LLMs (e.g., "in a note taking app you can create a new note with details"). Then, we ask crowdworkers to instantiate each feature description into one or multiple tasks involving apps of their choice.

By allowing annotators to use any app of their choice we succeed in collecting a largely-varied dataset encompassing 833 Android apps, including Google apps (Settings, Gmail, Google Maps, etc.), high-trend apps (e.g., Amazon, Booking.com, Kayak, Spotify, etc.) as well as less-popular or regional apps. This is important because high-popularity apps tend to include well-annotated accessibility trees and have more user-friendly interfaces, thus possibly facilitating the agent's task. We confirm this assumption by analyzing the performance of some of our tested agents on Google apps and non-Google apps (see results in Section E.5 in the Appendix).

During collection of a demonstration, annotators first provide a high-level description of a task in natural language (e.g., "Add an alarm to wake me up on Saturday mornings at 6am"). We ask annotators to make the descriptions detailed enough to be interpretable

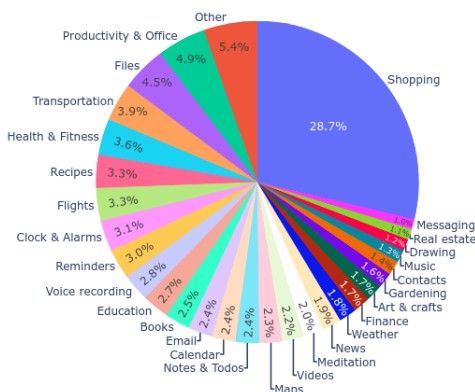

Figure 2: Distribution of the app categories that compose ANDROIDCONTROL.

without any ambiguity. We also instruct them to always include the name of the target app in the task description, unless obvious (e.g., Google first-party apps such as Clock or Settings). By doing so, the collected data can enable us to test memory-less, single-turn agent interactions.

In order to collect interaction traces, each annotator is provided with a setup that includes a physical Android phone (Google Pixel with Android 8.0 or higher) installed with a companion Android app that in turn connects to a web app running on a desktop Chrome browser. Annotators control the phone through the web app, using the WebUSB protocol and Android Debug Bridge (ADB). The web app provides annotators with controls to perform actions on the phone and observe their outcome. An annotator can select from the following set of UI actions to perform on the phone: click, long_press, input_text, scroll, navigate_home, navigate_back, open_app and wait (see Table 2). For each action, applicable metadata such as touch coordinates, target elements, entered text, and timing information

| | |
|---|---|
| click <elem> | Click the center of the specified element. |
| long_press <elem> | Long press the center of the specified element. |
| input_text <text> | Type the specified text. |
| scroll <direction> | Scroll in the specified direction. |
| navigate_home | Go to the home screen. |
| navigate_back | Go back to the previous app screen. |
| open_app <app> | Launch the specified app. |
| wait | Wait until the next observation is received. |

Table 2: Actions captured in ANDROIDCONTROL.

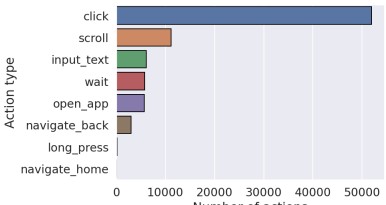

Figure 3: Action distribution

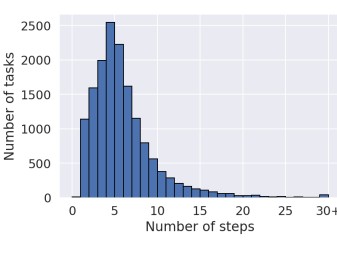

(a) Task length distribution

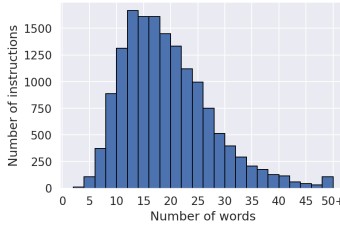

(b) HL instr. length distribution

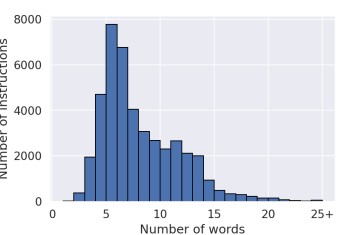

(c) LL instr. length distribution

Figure 4: Dataset statistics.

are automatically appended to the interaction trace (see Appendix B for more details). Annotators are instructed to avoid performing actions that are unnecessary or unrelated to the task. After an action is executed, a real-time screenshot of the phone's display is shown to the annotator and added to the interaction trace. This enables the annotator to completely operate their phone through the web app. Before executing each action, the annotator is asked to type in a short natural language description of the action they are about to take ("add a new alarm", "set the hours to 6", etc.), as if they were instructing someone to execute that action. These are also incorporated into the interaction trace and make up the low-level instructions in the dataset. If annotators realize the task is not feasible because of an unsupported functionality in the app or because of an error they tag the trace as infeasible or failed, respectively. Otherwise, it is tagged as successful.

As both high-level and low-level instructions are generated by annotators, they may contain typo and grammatically errors (for example, the high-level instruction in Figure 1). We left such errors in the dataset because humans often produce errors of this type.

Overall, this data collection involved 20 annotators. Each annotator went through a training process of several weeks. To maximize the diversity of the task demonstrations, in the last 4 months of the data collection we asked annotators to impersonate 20 different personas. Persona profiles are generated using an LLM prompted with detailed ranging from name, address, occupation, hobbies to upcoming week-end plans, family relationships, and typical day schedule.

### 3.2 Dataset statistics

Data statistics about ANDROIDCONTROL are summarized in Table 1. In addition, Figures 3 and 4 report distributions of UI actions, task lengths, and lengths of high and low level instructions. The task length distribution (Figure 4a), measured as number of steps required to complete the task, shows that tasks are of moderate length (between 1 and 13 steps for the 5th to 95th percentile, respectively). Lengths of high-level (HL) instructions fall between 8 and 34 words for the 5th and 95th percentile, and low-level (LL) instructions are between 3 and 14 words for the 5th and 95th percentile.

### 3.3 Dataset splits

We create a train, a validation and 4 test splits whose number of task demonstrations (episodes) and characteristics are detailed in Table 3. In order to measure how performance scales in domain and out of the domain of the collected data, we create the following test sub-splits: *1) in domain data (IDD)*: randomly pulled episodes from the same distribution as the training data; *2) app-unseen*: a test split using apps not present in the train split; *3) task-unseen*: a test split with tasks not present in the train

Table 3: Details on ANDROIDCONTROL train, validation and test splits. For every split, we report number of episodes, step actions, unique apps, app categories, and UI elements per screen.

| Split | Sub-splits | # Episodes | # Step actions | # Apps | # Categories | Avg. # elements per screen |
|-------|-----------|-----------|---------------|--------|--------------|----------------------------|
| Train | - | 13,604 | 74,722 | 769 | 39 | 222.2 |
| Val | - | 137 | 690 | 99 | 29 | 214.4 |
| Test | IDD | 721 | 3,897 | 296 | 35 | 221.5 |
| | App-unseen | 631 | 3,475 | 64 | 12 | 185.4 |
| | Task-unseen | 803 | 4,464 | 90 | 12 | 181.6 |
| | Category-unseen | 700 | 3,891 | 68 | 4 | 184.7 |

split; and *4) category-unseen*: a test split with apps from categories not present in the train split. Note that the test splits may contain overlapping episodes. For example, episodes in the unseen-category split will also be in the unseen-app and unseen-tasks splits. It is also the case that the average number of elements per screen can vary significantly between training and testing. Appendix B.4 explains why that is the case.

## 4  Experiments and results

In order to test the impact of data scale and task complexity on transfer performance in domain and out of domain, we conduct experiments in which we train on different amounts of the data in the ANDROIDCONTROL's training set. We also test zero-shot and few-shot methods.

### 4.1  Agent implementation

We implement a UI control agent for Android. The agent receives task instructions expressed in natural language. It observes the environment (the device) by deriving textual representations of the screen directly from the Android accessibility tree. The screen representation lists the on-screen UI elements. Each element is described according to the following attributes: type, text, content description, bounding boxes and various state tags (e.g., clickable, scrollable, focused, checked). As mobile UI screens may contain hundreds of UI elements (200 on average in ANDROIDCONTROL, Table 3), we pre-process the screen representations to include only UI elements that have a non-empty text description or UI elements of critical types (switch and edit). This process facilitates the agent's task and reduces the input's size. Note that our agent implementation does not directly leverage the page screenshot. While recent work explores how to infer screen representations from raw screens [18], best performance is still reported when using accessibility trees or HTML [38, 43, 28]. We expect the general trends we observe will hold true for multimodal language models.

During execution, the agent maintains a history over the previous steps. To avoid excessively large inputs, in the agent's input we include the screen description of only the current screen but append a history of the previously executed actions. In contrast to the action prediction output that locates UI elements by absolute coordinates, an action in the history is described in a self-contained manner, using its textual description and without any external reference.

The agent predicts an action among a set of candidate actions. The set of available actions matches the actions defined by ANDROIDCONTROL (Table 2) with two main changes. We add a terminate action that the agent predicts when it deems the task complete or infeasible. As this action is not originally provided in the dataset, for training purposes, we artificially insert it at the end of every episode (see Appendix D.1). For efficiency reasons, as in prior work [21], the input_text action is modified to include also the preceding click action necessary to set focus on the target element. The agent predicts the action type and any required arguments for the action, specifically the target UI element for a click action, the text to be typed and the target element for a typing action, the name of an app to open, the direction of a scroll, etc. For an example of screen representation, a summary of the agent's action space, and more details on the agent implementation please refer to Appendix C.

### 4.2  Experimental setup

The LLMs we experiment with include PaLM-2L [9], PaLM-2S [9], Gemini 1.5 Pro [8], GPT-4 and GPT-4 Turbo [26]. We set the temperature to zero for all models to obtain more deterministic

responses. To limit compute, we perform fine-tuning only with PaLM-2S, and adopt the parameter efficient tuning approach of LoRA [14]. We set the LoRA rank to 4 when fine-tuning with small amounts of data ($<$ 1k episodes), while switch to a rank of 64 when using more episodes. For few-shot experiments we use Gemini 1.5 Pro which provides a context window of 1M tokens.

We create SeqIO [29] tasks to extract data points from ANDROIDCONTROL and to generate prompts and target outputs (more details in Appendix D.1). We setup two SeqIO tasks: (i) SeqIO HL (high-level) where only a high-level instruction is included in the prompt, and (ii) SeqIO LL (low-level) where both a low-level instruction and its corresponding high-level instruction are included. This second task emulates the use case where an LLM is used for decomposing high-level instructions into a sequence of low-level commands and another LLM is used for grounding; the assumption is that this grounding LLM may improve performance by having access to the context of the high-level command. In addition to the natural language instruction(s), each data point contains the textual description of the start screen, the history of performed actions, and the ground-truth action. Through these two SeqIO tasks, we investigate how a model performs on simpler (LL) or harder (HL) task instructions.

To reduce LLM costs, some zero-shot and all few-shot evaluations are done on a subset of the test split of ANDROIDCONTROL, *Random-500*, that contains 500 random step actions from the full test split and has a similar sub-split distribution. We verified through experiments that results on Random-500 are a good approximation of the results on the full test split (Appendix E.3).

**Zero-shot**    We test four zero-shot methods. (i) We use the AitW [27] prompt, specifically designed for Android and the PaLM model, without any modifications. (ii) We adapt the best-performing SeeAct [43] variant ("choice") which grounds actions via textual choices. SeeAct was originally designed for GPT-4V for web navigation tasks. At each step, SeeAct queries the LLM twice. In the first query, it analyzes the state and performs reasoning. Then, in the second query, it asks the LLM to select an action from multiple choices. We use the SeeAct prompt by Rawles et al. [28] adapted to work on mobile and to take textual representations of Android screens as input. (iii) We evaluate the text-only version of M3A [28] that combines ReAct-style [42] and Reflexion-style [32] prompting. (iv) Finally, we test a zero-shot prompt (implementation in Appendix D.2.2) of the same form we use with our agent described above. This allows us to measure performance of a base model for our agent without any fine-tuning. This prompt emphasizes the use of a screen description composed of UI elements, hence the name ER (Element Representations). Note that with the exception of ER, which we ran with all 4 base models, to limit prompt sensitivity [30], we ran the other prompts with the model family they were originally designed for.

**Few-shot and LoRA-tuned models**    When evaluating few-shot on HL instructions, samples drawn from the HL SeqIO task are used in the prompt. When testing on LL instructions, samples from the LL SeqIO task are included. For convenience, LoRA-tuned models are trained on a mixture of both the LL and HL SeqIO tasks as we found training on the two SeqIO tasks separately or in a mixture to achieve similar accuracy (see Appendix E.2). Best model checkpoints are selected using the validation split. We use the simple ER prompt for few-shot and LoRA-tuned models.

**Scaling analysis**    To conduct the scaling analysis, we vary the number of samples included in the prompt of few-shot (FS) techniques or in the training set of LoRA-tuned (LT) models. We randomly sample episodes from the SeqIO tasks using the following sample sizes: 5, 10, 100, 1k, 10k, and all (13,604) episodes. For few-shot only, to make the prompts more varied, we sample an equivalent number of step-examples from different episodes.

**Metrics**    As in prior work [43, 27], as our evaluation metric we adopt step-wise accuracy, which measures the success of each task step. A step is successful if the predicted action and arguments (target element and text, if present) are correct. We adopt a relaxed metric that considers equivalent actions in additions to exact matches as successful (see Appendix D.3 for details).

### 4.3   In-domain performance

We start by evaluating zero-shot, few-shot and LoRA-tuned methods in domain. Table 4 reports the step-wise accuracy performance on the IDD sub-split of Random-500. In-domain, LoRA-tuned models, despite using the smaller PaLM 2S model, when trained with sufficient amounts of data, largely outperform the zero-shot and few-shot methods. For low-level instructions, even LT-5

Table 4: Performance on the IDD sub-split of Random-500. For few-shot (FS) and LoRA-tuned (LT) methods -$X$ (-$all$) indicates $X$ ($all$) episodes are used in the prompt or in training. Unless noted with a "r64" for models fine-tuned with a LoRA rank of 64, LoRA-tuned models use rank=4.

| Regime | Method | Model | Step accuracy | |
|---|---|---|---|---|
| | | | high-level instr. | low-level instr. |
| Zero-shot | AitW | PaLM 2L | 19.5 | **56.7** |
| | SeeAct | GPT-4-Turbo | 33.9 | 54.3 |
| | M3A | GPT-4-Turbo | **42.1** | 55.0 |
| | ER | PaLM 2S | 19.5 | 45.5 |
| | ER | PaLM 2L | 33.0 | 45.9 |
| | ER | GPT-4 | 32.1 | 51.7 |
| | ER | Gemini 1.5 Pro | 24.4 | 50.2 |
| Few-shot | FS-5 | Gemini 1.5 Pro | **41.8** | 50.2 |
| | FS-10 | Gemini 1.5 Pro | 40.2 | 50.8 |
| | FS-100 | Gemini 1.5 Pro | 39.5 | **53.3** |
| LoRA-tuned | LT-5 | PaLM 2S | 30.3 | 57.1 |
| | LT-10 | PaLM 2S | 28.5 | 58.9 |
| | LT-100 | PaLM 2S | 39.8 | 62.8 |
| | LT-1k | PaLM 2S | 52.5 | 71.4 |
| | LT-10k | PaLM 2S | 62.0 | 85.7 |
| | LT-all | PaLM 2S | 65.6 | 81.8 |
| | LT-1k-r64 | PaLM 2S | 54.8 | 76.6 |
| | LT-10k-r64 | PaLM 2S | 69.6 | 81.9 |
| | LT-all-r64 | PaLM 2S | **71.5** | **86.6** |

surpasses all non-fine-tuned models, while for high-level instructions, it requires more training data (1k episodes). The best fine-tuned model reaches 71.5% on high-level instructions and 86.6% on low-level instructions.

The best zero-shot performance on low-level instructions is obtained with AitW using PaLM 2L (56.7%) and on high-level instructions with M3A using GPT-4 (42.1%). This performance likely reflects the design of the prompts, the strength of the different base models used and the benefits of incorporating some high-level reasoning (included in M3A) for handling high-level instructions. Interestingly, the few-shot performance is for the most part inferior to that of zero-shot methods.

## 4.4 Effect of scale on in-domain transfer

Fine-tuning obtains good performance in domain, but how much data is needed for acceptable performance? A failure at a single step may prevent task completion, a phenomenon we refer to as the "weakest link effect." Making the simplifying assumption of i.i.d. success across steps, completing, for example, a 5-step task correctly 95% of the time requires a 99% step-wise accuracy. We perform an analysis to extrapolate the amount of training data required to achieve such performance.

Figure 5a visualizes the number of training episodes drawn from ANDROIDCONTROL and the step accuracy achieved on the full IDD test sub-split. Both high and low-level curves exhibit linear trends ($R^2$ coefficients are greater than 0.95) with the log of training data. We extrapolate that it would take 500K and 1M episodes to reach 95% step-accuracy for low and high-level instructions, respectively. However, while low-level tasks can be accomplished in one step, high-level tasks require multiple steps. Taking 5 steps as the rough length for high-level tasks, we predict 2M episodes would be required to reach the 99% step-wise accuracy to achieve 95% episode completion for high-level tasks (see Appendix E.4 for an empirical evaluation of performance as episode length is varied). While this is conservative by assuming no possible mistake recovery, we still feel this analysis provides a helpful rough quantification.

## 4.5 Effect of scale on out-of-domain transfer

We now use ANDROIDCONTROL's out-of-domain test splits (Table 3) to quantify how fine tuning with more demonstrations affects out-of-domain performance. This is important for assessing the robustness of agents used in the real world on tasks not foreseen in the data used to train an agent.

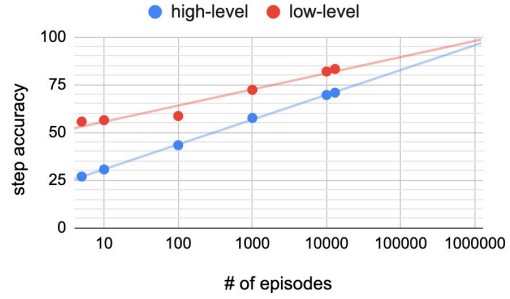

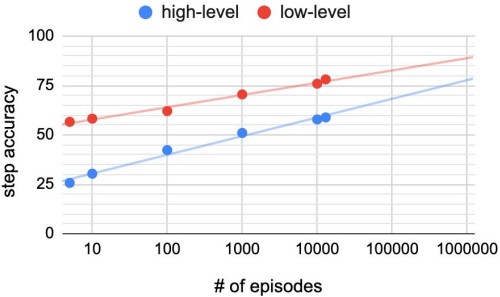

(a) In-domain. High-level trendline $R^2 = 0.999$, low-level trendline $R^2 = 0.951$.

(b) Out-of-domain. High-level trendline $R^2 = 0.986$, low-level trendline $R^2 = 0.987$.

Figure 5: Relationship between step-wise accuracy and number of fine-tuning samples. With 5–100 episodes we use LoRA rank=4; with $\geq$1k episodes we use LoRA rank=64. IDD performance is based on the full IDD test split. OOD performance is based on the average across the three OOD splits.

Table 5: OOD step accuracy. In square brackets [X] we report the percentage point increase (+) or drop (-) from the IDD accuracy obtained on the full IDD test split.

|  |  | IDD | app-unseen | task-unseen | categ-unseen |
|---|---|---|---|---|---|
| LT-5 | HL | 26.9 | 25.7 [-1.2] | 26.4 [-0.5] | 25.1 [-1.8] |
|  | LL | 55.7 | 56.9 [+1.2] | 56.6 [+0.9] | 56.4 [+0.7] |
| LT-10 | HL | 30.6 | 29.9 [-0.7] | 31.1 [+0.5] | 30.2 [-0.4] |
|  | LL | 56.4 | 58.3 [+1.9] | 58.2 [+1.8] | 58.2 [+1.8] |
| LT-100 | HL | 43.3 | 42.4 [-0.9] | 42.5 [-0.8] | 42.1 [-1.2] |
|  | LL | 58.6 | 62.7 [+4.1] | 61.7 [+3.1] | 61.8 [+3.2] |
| LT-1k | HL | 53.2 | 49.0 [-4.2] | 49.3 [-3.9] | 48.1 [-5.1] |
|  | LL | 68.0 | 68.0 [ 0.0] | 67.3 [-0.7] | 67.4 [-0.6] |
| LT-10k | HL | 63.9 | 55.2 [-8.7] | 55.6 [-8.3] | 54.2 [-7.7] |
|  | LL | 78.7 | 76.7 [-2.0] | 75.6 [-3.1] | 75.5 [-3.2] |
| LT-all | HL | 65.5 | 58.7 [-6.8] | 59.7 [-5.8] | 58.2 [-7.3] |
|  | LL | 80.7 | 78.6 [-2.1] | 77.9 [-2.8] | 77.8 [-2.9] |
| LT-1k-r64 | HL | 57.6 | 51.1 [-6.5] | 51.7 [-5.9] | 50.2 [-7.4] |
|  | LL | 72.3 | 71.0 [-1.3] | 70.4 [-1.9] | 70.1 [-2.2] |
| LT-10k-r64 | HL | 69.6 | 57.7 [-11.9] | 56.9 [-12.7] | 58.9 [-10.7] |
|  | LL | 81.9 | 76.3 [-5.6] | 75.8 [-6.1] | 75.2 [-6.7] |
| LT-all-r64 | HL | 70.8 | 58.5 [-12.3] | 59.6 [-11.2] | 57.4 [-13.4] |
|  | LL | 83.2 | 78.5 [-4.7] | 77.3 [-5.9] | 76.8 [-6.4] |

As the number of fine-tuning samples increases, performance improves and so does the gap between IDD and OOD performance (Table 5). With 10k or more episodes, the IDD accuracy is noticeably higher than on the three ODD splits. For example, with LT-10k (r=4), the gap is 7.7–8.7 pp for high-level instructions and 2.0–3.2 pp for low-level instructions. With r=64 the gap increases. In general, more out-of-domain transfer occurs for low-level tasks, which is expected as low-level tasks share more similarity across tasks and apps than high-level tasks.

As for in-domain, we extrapolate how much training data would be necessary to achieve a reasonable accuracy out of domain (Figure 5b). OOD step-accuracy grows more slowly than in domain, and is estimated to reach 95% at 10M and 60M episodes for low-level and high-level instructions, respectively. Similar to above, the number of episodes we predict would be required to reach 99% step accuracy to therefore achieve 95% episode completion rate on 5-step high-level tasks is 150M. Based on these projections, it seems expensive but feasible to obtain good general LL performance with fine-tuning, while the predicted higher order of magnitude of the number of required demonstrations suggests fine-tuning alone may not be sufficient to achieve robust OOD performance on HL tasks.

**More experiments.** To complete our evaluation analysis in Appendix E we run more experiments studying the impact of other factors, including episode length, action types, and app types.

## 5  Limitations

There are multiple potential limitations in this work. First, we only fine-tuned one model, PaLM-2S; however, while the absolute performance values would change, we expect our relative findings to be consistent across model families. Additionally, using offline evaluation for agent performance has the known issue of not rewarding alternative routes to complete a task and the ability to take corrective actions [3, 27]. Further, while selected to encompass important use cases, the set of app categories in ANDROIDCONTROL is still an incomplete representation of all tasks users may ask agents to perform. Finally, studying inference costs is not a focus of the paper, although it is obvious that it is much cheaper to predict on a fine-tuned PaLM-2S model than on any larger models, such as PaLM-2L or GPT-4.

## 6  Conclusion

We have introduced ANDROIDCONTROL, a large and diverse dataset structured for studying the performance of models in and out of domain on low and high-level tasks, as training data is scaled. Using this dataset, we evaluate scaling of LoRA fine-tuned models. We predict that to achieve 95% accuracy for in-domain low-level tasks, 1M episodes would be required, while 2M episodes would be required to obtain 95% episode completion rates for 5-step high-level tasks. While these results are for only one model, they suggest that fine-tuning may be a viable, though possibly expensive, route for obtaining high in-domain performance for low and high level tasks. Out of domain, 10M and 150M episodes would be required, respectively. This one to two orders of magnitude increase suggests fine-tuning may not scale well out of domain, and may not be sufficient to obtain good out-of-domain performance on HL tasks.

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

# A  Ethical considerations

Autonomous UI control agents can bring value to visually-impaired users, by providing them with access to a much wider range of applications and functionality. More broadly, they can enhance human productivity by automating everyday tasks. UI agents have societal, security and privacy implications. An agent may leak private information or carry out a task in an unacceptable way or produce unwanted side effects. Malicious actors could also use these agents for undesired purposes such as overriding anti-fraud mechanisms or manipulating applications to achieve undesirable goals. For these reasons, deployment of this technology going forward will have to be carefully considered and combined with research in other areas on LLM safety to balance potential societal trade-offs with risks.

In our experiments, we used the PaLM 2 model, which is available publicly through the Vertex AI PaLM API from Google. Our research use was in accordance with Google's AI prohibited use policy (`https://policies.google.com/terms/generative-ai/use-policy`).

# B  Dataset details

## B.1  Data collection

The data collection was carried out by annotators who are paid contractors, who received a standard contracted wage, which complies with living wage laws in their country of employment. The annotators were informed of the intended use of the data collected and signed a data usage agreement. They did not use their personal devices nor they were required to enter any private information.

We provided annotators with a detailed instructional document and video tutorials on how to operate the Android and web apps for data collection. All raters went through a training phase where they could familiarize with the tools and received personalized feedback based on manual inspection of the collected traces.

Examples of episodes from ANDROIDCONTROL are shown in Figure 6.

## B.2  Dataset format

ANDROIDCONTROL is publicly released at `https://github.com/google-research/google-research/tree/master/android_control`. Each datapoint is stored as a TFRecord file with the following fields:

- *episode_id*: a unique identifier integer for each episode. This is especially useful when generating the data splits.
- *goal*: the high-level instruction for the entire episode.
- *screenshots*: a list of screenshot byte strings for each observation encoded as PNGs.
- *accessibility_trees*: a list of Android accessibility trees for each observation.
- *screenshot_widths*: a list of the widths of each of the screenshots.
- *screenshot_heights*: a list of the heights of each of the screenshots.
- *actions*: a list of actions represented as JSON dictionaries. The actions are performed between consecutive screenshots, so there are len(screenshots) - 1 of them.
- *step_instructions*: a list of the low-level instructions describing each step to complete the task. The number of step instructions equals the number of actions, but it is important to note that each step instruction does not necessarily describe a single action. A step instruction can require more than one action to complete, and in these cases the step instruction is repeated to maintain a one-to-one mapping from step instructions to actions.

## B.3  Accessibility node metadata

Each node in the Android accessibility tree corresponds to a UI element in the screen. Each node is described by multiple metadata. In our UI control agent implementation we use the following node metadata:

Share my favorite Book "the Queen's Gambit" to my Friend Natalie larson over her gmail address -natalie.larson1998@gmail.com from the PocketBook app.

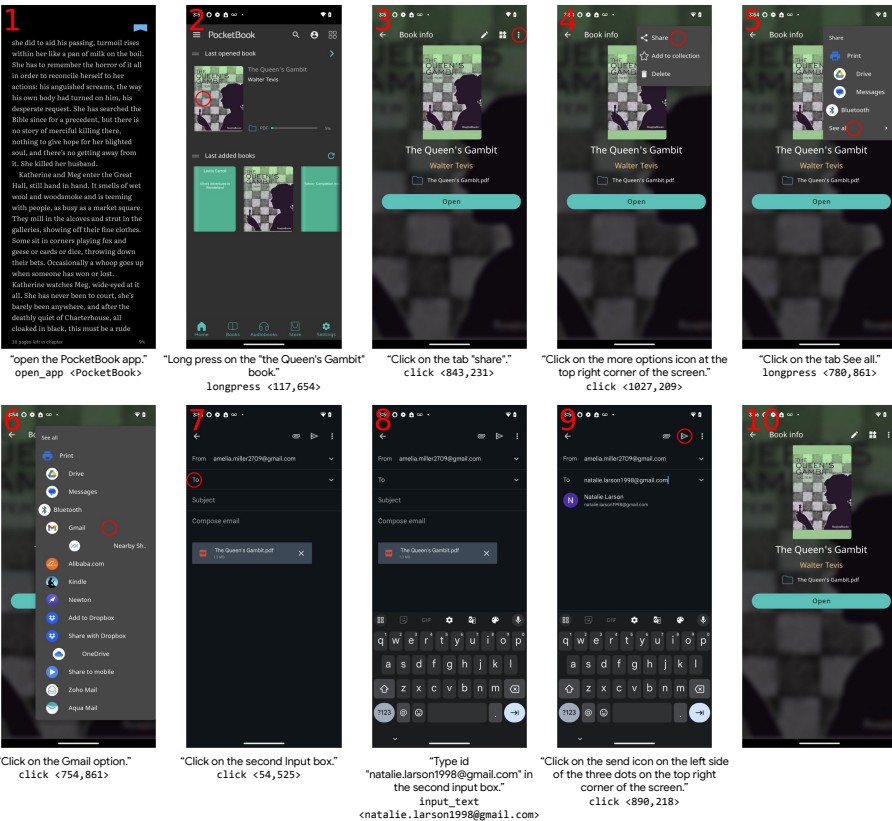

Launch the Amazon app, search for India gate basmati rice, and add 5 Kg of it to your shopping cart.

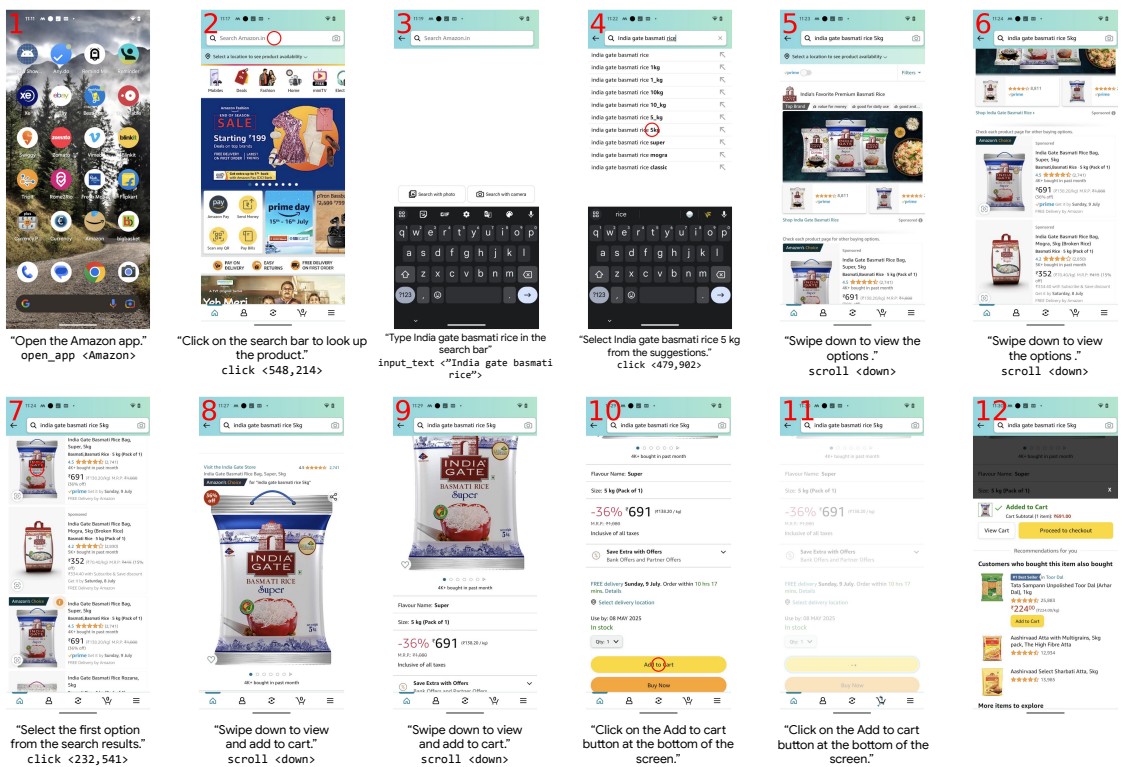

Figure 6: Example episodes contained in ANDROIDCONTROL. Red circles highlight the location of click and long press actions on the screen. Red numbers are added only for illustration purposes.

- Element type: class_name (e.g. Button, TextView, Image, etc.)

- Textual attributes: text, content_description, hint_text, tooltip_text, view_id_resource_name.

- Location and size: bounds_in_screen.

- Element status (as boolean): is_checked, is_enabled, is_focused, is_selected.

- Element properties (as boolean): is_checkable, is_clickable, is_editable, is_focusable, is_long_clickable, is_scrollable, is_password, is_visible_to_user.

## B.4  Number of UI elements in test splits

In Table 3, we note that in the test sub-split that is sampled from the same distribution as the train split (the "IDD" split), the average number of UI elements per screen is comparable, as expected, while in the other 3 out-of-domain test sub-splits there is a significant drop in the number of UI elements per screen. To explain this phenomenon, we calculated the average number of UI elements across all screens in the data for each app (for simplicity we call this "the average number of UI elements per app"). Figure 7 shows the distribution of the average number of UI elements per app for the entire dataset. The distribution shows a long tail, with a few apps with a very large number of UI elements per screen. If we remove these outliers by dropping the top 5% of apps with the most elements per screen, the average number of UI elements across the remaining apps is 180.9, which is inline with the out-of-domain test sub-splits. A reasonable question is then why do the out-of-domain test sub-splits not sample from the tail? We believe this is because, as shown in Table 3, the number of apps in each sub-split is relatively small. Further, there is overlap between the different test sub-splits (this was done to keep the total size of the test set reasonable, if each sub-split needed to be disjoint, it would leave less train data), so the total number of apps across the union of all out-of-domain test sub-splits is still modest, again supporting that we simply did not draw enough outliner apps from the long tail to affect the average for the out-of-domain splits.

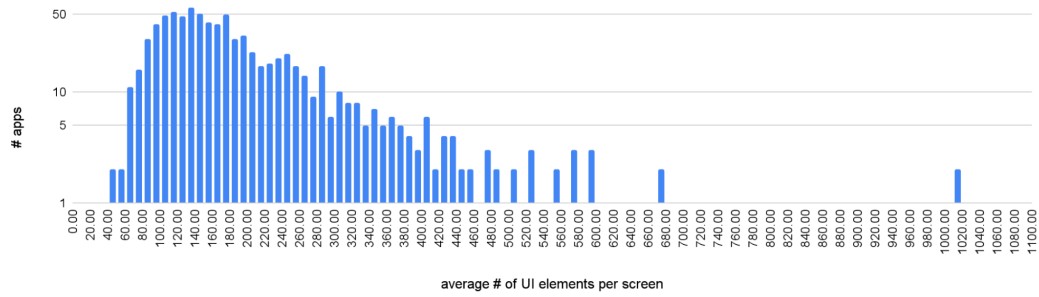

Figure 7: Histogram of average number of UI elements per app for all apps in the ANDROIDCONTROL dataset. Please note the y-axis is logarithmic.

# C   UI control agent implementation

## C.1  Observation space

The device state is perceived through the UI screen currently displayed. A screen representation is derived from the Android accessibility tree and lists all UI elements composing the UI. Each element is described by three fields: text (a textual description derived from the element's textual description, content description, class name, etc.), position, and status (e.g., whether a checkbox is selected). These fields are populated using the metadata (or a combination thereof) associated with Android accessibility nodes (Appendix B.3). For simplicity, in this paper, we only experiment with screen descriptions that consist of a flat list of UI elements. Figure 8 shows an example screenshot and the corresponding JSON screen representation.

Table 6: Agent's action space. For each action the table reports the JSON template the agent is asked to predict.

| Action | JSON template |
|---|---|
| click | {"action_type":"click","x":<x_coordinate>,"y":<y_coordinate>} |
| long_press | {"action_type":"long_press","x":<x_coordinate>,"y":<y_coordinate>} |
| type | {"action_type":"type","text":<text_input>,"x":<x_coordinate>,"y":<y_coordinate>} |
| scroll | {"action_type":"scroll","direction":<up, down, left, or right>} |
| navigate_home | {"action_type":"navigate_home"} |
| navigate_back | {"action_type":"navigate_back"} |
| open_app | {"action_type":"open_app","app_name":<app_name>} |
| wait | {"action_type":"wait"} |
| terminate | {"action_type":"status","goal_status":<"successful","infeasible">} |

## C.2 Action space

When predicting an action that involves a target element, the model should output sufficient details to locate the target UI element, either an index or its geometric information such as its bounding rectangle. To reduce the complexity of parsing the model output, we prompt an LLM to output its action selection in a predefined JSON format. In the case of element click, for instance, the model outputs a prediction in the following format: {"action_type":"click","x":<x_coordinate>,"y":<y_coordinate>}, where the target element is identified by its center coordinates. We found LLMs work equally well with predicting element centers or element indices, but as the former approach is compatible with click actions that are not restricted to specific UI elements, our implementation always outputs the center of the target UI element. The same applies to all actions that take an element as input.

Table 6 lists the JSON action templates and defines the agent's action space. Compared to the actions collected in ANDROIDCONTROL (Table 2), there are two main differences. First, we introduce the type action derived from the dataset's input_text action. This action is obtained by aggregating the input_text action and its preceeding click action, which is necessary to focus on the UI element before typing. Accordingly, the low-level task instructions are merged via concatenation. This unified action is more efficient from an agent implementation's perspective and reduces latency at execution time. Second, we introduce a new action, terminate, which signals whether the agent deems the task as successfully completed or infeasible. To support training and testing of this action, we insert an additional step at the end of each episode with low-level instruction "terminate', action=terminate, and value set to "successful" or "infeasible" depending on the episode status.

## C.3 History

Actions performed in previous steps and their outcome are included as the history of the current step. An action in history is derived from the JSON action so that it is self-contained without any external reference. Section C.4 contains an example of history. Please note that in an offline dataset, as in this paper, the outcome of a previous action is recorded by annotators and most likely successful. However, in a real system, the underlying framework can report an action as failed due to screen synchronization errors or prediction errors which may render a target element not localizable or an action not executable.

## C.4 Examples of screen description, JSON action, and history

Figure 8 shows an example screenshot annotated with UI elements, and the list next to it is the corresponding screen description. The position and shape of each UI element are defined by "center" and "size" while its semantic meaning is described by the "text" field. Note that a switch element does not have any textual attribute, therefore a text label "Switch" derived from its class_name is assigned (text in red), and its status is specified by the "checked" field (text in blue).

Given a goal example : "search for lord of the rings". The ground truth output is an action in JSON format, "action_type":"click","x":539,"y":2078 , where (539, 2078) is the center of the search bar at the bottom.

The following is an example of action history that is included in the prompt after two actions:

```
{"UI elements":[{"center":[350,1601],"size":[323,77],"
    text":"Clear history"},{"center":[128,1491],"size
    ":[131,60],"text":"Display"},{"center
    ":[158,1420],"size":[190,81],"text":"Display"},{"
    center":[320,1265],"size":[514,59],"text":"
    Location > Location services"},{"center
    ":[317,1194],"size":[509,82],"text":"Bluetooth
    scanning"},{"center":[128,1038],"size":[131,60],"
    text":"Display"},{"center":[271,967],"size
    ":[416,81],"text":"Brightness level"},{"center
    ":[943,772],"size":[188,225],"text":"Switch",
    "checked":true},{"center":[497,812],"size
    ":[616,60],"text":"Connected devices > Connection
    "},{"center":[317,741],"size":[256,81],"text":"
    Bluetooth"},{"center":[94,772],"size":[63,225],"
    text":""},{"center":[943,546],"size":[188,225],
    "text":"Switch","checked":true},{"center
    ":[449,586],"size":[520,59],"text":"Network &
    internet > Internet"},{"center":[257,516],"size
    ":[136,81],"text":"Wi-Fi"},{"center":[94,546],"
    size":[63,225],"text":""},{"center":[566,370],"
    size":[1027,126],"text":"RECENT SEARCH RESULTS
    "},{"center":[603,219],"size":[953,126],"text":"
    Search settings"},{"center":[63,219],"size
    ":[126,147],"text":"Back"},{"center":[996,68],"
    size":[19,136],"text":"Battery 100 percent."},{"
    center":[950,67],"size":[38,38],"text":""},{"
    center":[892,67],"size":[78,41],"text":"No
    internet"},{"center":[348,68],"size":[57,136],"
    text":"Android System notification: C"},{"center
    ":[290,68],"size":[58,136],"text":"Android System
    notification: S"},{"center":[209,68],"size
    ":[105,136],"text":"09:22"}]}
```

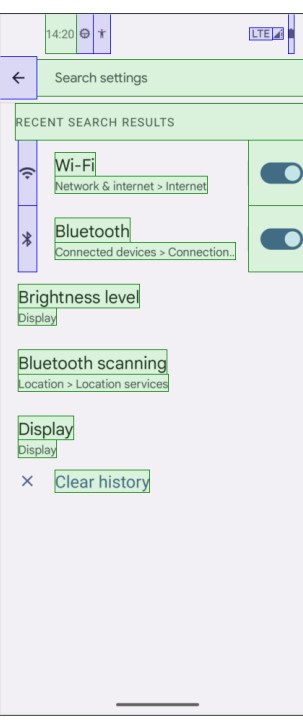

Figure 8: An example screenshot annotated with UI elements (right) and corresponding screen representation (left). Note that this example is picked purposely to have few UI elements so that the screen description is compact.

```
{"0":["click [Search]","successful"], "1":["type "lord of the rings" at [Search apps
    , web and more]","successful"]}
```

Note that target elements, such as [Search] and [Search apps, web and more], are identified by their text labels or descriptions, hence do not reference the corresponding screen description.

# D    Experimental details

## D.1    Data processing and training details

To run our experiments we generate 2 SeqIO tasks (HL and LL) that process the ANDROIDCONTROL dataset as follows. (i) In order to support prediction of task completion actions not present in the original dataset, at the end of every episode we artificially insert a terminate action that takes the episode status (successful or infeasible) as argument. (ii) The dataset omits identifiers for target elements associated with click or long_press actions. Such identifiers can be retrieved by determining which accessibility node encompasses the click location. When multiple nodes satisfy this criterion, the node with the smallest area is selected. However, we discard any step with an element-based action (click, long_press) that does not have a UI element associated. This is due to either a touch on an empty area or a target UI element missing from the accessibility tree. These discarded steps are still considered in the action history to support prediction of later steps that reference previous actions or elements. (iii) Finally, we discard (only from SeqIO LL tasks) steps that are missing a low-level

Table 7: Statistics of the SeqIO tasks generated from ANDROIDCONTROL. Note that the number of steps here reported is different than what reported in Table 3 because it accounts for terminate and type actions which are not present in the original dataset (see C.2).

| Split | SeqIO tasks | # Episodes | # Steps | Avg # steps per episode |
|---|---|---|---|---|
| Train | HL | 13,604 | 70,796 | 5.2 |
| | LL | 13,344 | 60,954 | 4.6 |
| Validation | HL | 137 | 655 | 4.8 |
| | LL | 135 | 559 | 4.1 |
| Test | HL | 7,897 | 1,543 | 5.1 |
| | LL | 6,833 | 1,500 | 4.6 |

instruction which annotators may have forgotten to enter. An episode is dropped only if all of its steps are discarded. Table 7 shows the statistics of the SeqIO tasks that are the results of this processing.

## D.2  LLM prompts for zero-shot experiments

For an LLM-based UI control agent, its prompt describes what the agent is expected to do, the action space and the expected format of actions, the task instruction, the current screen and application, and the history of previously executed actions and their outcome.

In our experiments we test four different prompts. We use the original AitW and M3A prompt as described in the original publications [27, 28]. For the SeeAct [43] and ER prompts we list the detailed prompts in the following. Please note that variable placeholders that are replaced by real values are marked by {{{ }}}.

### D.2.1  SeeAct prompt

We took the Android adaptation of the SeeAct [43] prompt by Rawles et al. [28]. As it is designed for GPT-4V, we slightly modify it by replacing its annotated pixel input by using the same JSON screen description as shown in Section C.1. We find that by using our element filtering approach fewer than 50 elements are usually present per screen. As a result, the ranker model of SeeAct which selects candidate UI elements is not necessary, hence it is disabled.

### D.2.2  ER prompt

We also design a new prompt, ER, which emphasizes the use of a screen description composed of UI elements. The prompt does not encourage an LLM to reason, hence it is much simpler than the other prompts. This prompt is also used for few-shot and fine-tuning experiments. The prompt is shown below:

```
An agent follows instructions on an Android device. Each instruction requires one or
    more steps. At each step, the input includes previous_actions, active_app,
    screen_width_and_height, and screen_description. You are required to select one
    action from the available actions.

# Available actions:
{"action_type":"click","x":<x_coordinate>,"y":<y_coordinate>}
{"action_type":"type","text":<text_input>,"x":<x_coordinate>,"y":<y_coordinate>}
{"action_type":"navigate_home"}
{"action_type":"navigate_back"}
{"action_type":"scroll","direction":<up, down, left, or right>}
{"action_type":"open_app","app_name":<app_name>}
{"action_type":"wait"}
{"action_type":"dismiss","x":<x_coordinate>,"y":<y_coordinate>}
{"action_type":"long_press","x":<x_coordinate>,"y":<y_coordinate>}
{"action_type":"get_text","x":<x_coordinate>,"y":<y_coordinate>}

If the goal of an instruction is reached, output the following special action
{"action_type":"status","goal_status":"successful"}
If the goal of an instruction is not possible, output the following special action
```

```
{"action_type":"status","goal_status":"infeasible"}

# Explanation of inputs
The top edge of a screen has y_coordinate equal to 0. The y_coordinate of the bottom
     edge of a screen equals to screen height. In screen_description, missing the '
    checked' field for an element indicates that it is NOT checked.
The size of an element is defined by width and height.

# Input
instruction: {{{grounding_goal}}}

previous_actions: {{{previous_actions}}}
active_app: {{{active_app}}}
screen_width_height: {{{screen_width,screen_height}}}

screen_description: {{{screen_description}}}

The action to take:
```

### D.2.3  Few-shot prompt

For a few-shot prompt, the following is inserted before "# Input" in the ER prompt:

```
Following are a few exemplars. Each exemplar is marked by <EXEMPLAR_i> and </
    EXEMPLAR_i> tags.
{{{exemplars}}}
```

The layout of an exemplar is as follows:

```
<EXEMPLAR_{{{exemplar_index}}}>
# Input
instruction: {{{grounding_goal}}}

previous_actions: {{{previous_actions}}}
active_app: {{{active_app}}}
screen_width_height: {{{screen_width,screen_height}}}

screen_description: {{{screen_description}}}

The action to take:
{{{ground_truth_action}}}
</EXEMPLAR_{{{exemplar_index}}}>
```

### D.3  Action matching

In computing step-wise accuracy, we consider an action correctly predicted if it matches the ground truth action exactly (i.e., action type and arguments are identical) or if it aligns with it as described below.

For element-based actions (click, long press, type), if the target element's coordinates are within the bounding box of the ground truth target element, it is considered as matching. This relaxation matches the behavior on Android devices where a touch gesture will activate an element as long as it falls within the element's bounds.

On Android the behaviour of the navigate_back action is equivalent to clicking on the on-screen "Back" button so we consider them equivalent. Similarly, open_app is considered equivalent to clicking a UI element whose text matches the app name.

# E    More experimental results

## E.1    Confusion matrices for action predictions

Table 8 and 9 show the (normalized) confusion matrices with regard to action type predictions obtained with the LoRA-tuned PaLM 2S model (trained on the entire dataset). Results are averaged across all four ANDROIDCONTROL's test splits. All the numbers are percentage.

Table 8: Confusion matrix for the action predictions of the LoRA-tuned PaLM 2S model (rank=64, trained on the entire training set) on high-level instructions.

| | | | | | Predicted | | | | |
| | | click | input_text | long_press | navigate_back | open_app | scroll | wait | terminate |
|---|---|---|---|---|---|---|---|---|---|
| | click | **88.1** | 0.6 | 0.1 | 1.5 | 0.0 | 4.6 | 3.8 | 1.3 |
| | input_text | 17.3 | **76.0** | 0.0 | 0.0 | 0.0 | 1.0 | 4.8 | 1.0 |
| | long_press | 28.6 | 0.0 | **71.4** | 0.0 | 0.0 | 0.0 | 0.0 | 0.0 |
| Actual | navigate_back | 23.9 | 0.0 | 0.0 | **64.9** | 2.9 | 6.7 | 1.6 | 0.0 |
| | open_app | 2.9 | 0.0 | 0.0 | 2.6 | **93.3** | 0.6 | 0.6 | 0.0 |
| | scroll | 17.9 | 0.1 | 0.0 | 1.5 | 0.0 | **70.9** | 4.5 | 5.1 |
| | wait | 14.1 | 0.3 | 0.0 | 1.6 | 0.0 | 10.8 | **68.6** | 4.6 |
| | terminate | 16.0 | 0.6 | 0.0 | 1.5 | 0.3 | 30.7 | 12.2 | **38.6** |

Table 9: Confusion matrix of actions predictions of the LoRA-tuned PaLM 2S model (rank=64, trained on the all training set) on low-level instructions.

| | | | | | Predicted | | | | |
| | | click | input_text | long_press | navigate_back | open_app | scroll | wait | terminate |
|---|---|---|---|---|---|---|---|---|---|
| | click | **93.3** | 0.3 | 0.0 | 0.2 | 0.0 | 0.8 | 3.7 | 1.7 |
| | input_text | 11.1 | **84.8** | 0.0 | 0.0 | 0.0 | 1.0 | 3.0 | 0.0 |
| | long_press | 12.5 | 12.5 | **75.0** | 0.0 | 0.0 | 0.0 | 0.0 | 0.0 |
| Actual | navigate_back | 2.3 | 0.0 | 0.0 | **93.7** | 0.3 | 0.0 | 3.2 | 0.6 |
| | open_app | 0.2 | 0.0 | 0.0 | 0.0 | **98.5** | 0.0 | 1.3 | 0.0 |
| | scroll | 3.3 | 0.1 | 0.0 | 0.1 | 0.0 | **90.4** | 2.0 | 4.1 |
| | wait | 11.8 | 0.2 | 0.2 | 1.4 | 0.0 | 4.6 | **76.0** | 5.8 |
| | terminate | 10.5 | 1.3 | 0.0 | 0.8 | 0.6 | 7.0 | 12.4 | **67.5** |

Actions of type click and open_app are predicted with high accuracy (above 88%). When UI actions are inferred from low-level instructions, performance is generally higher and mispredictions mainly occur for long_press and terminate actions. Long press actions are not common in the dataset and in general in mobile apps, hence the model does not learn them as well as other actions. Task completion is generally hard to learn for the model and in many cases it is wrongly recognized as a pause. When UI actions must be inferred from high-level instructions, performance is naturally lower as the model must decompose the high-level instruction into a sequence of lower-level actions, thus requiring decision making and reasoning capabilities. The most challenging actions are terminate, navigate_back, and wait. These actions are generally not explicit in a user instruction (e.g., a user may say "download the file" rather than "download the file and wait for the download"), therefore requiring further reasoning and pre-knowledge of the task flow.

Please note that Table 8 and 9 do not consider action arguments. Table 10 shows the accuracy of predicting all action arguments when the action type is predicted correctly.

Table 10: Accuracy of predicting action arguments when the action type is predicted correctly. The numbers are in percentage. Results obtained with the LoRA-tuned PaLM 2S model (rank=64, trained on the all training set)

| action type | click | input_text | long_press | open_app | scroll |
|---|---|---|---|---|---|
| high-level instructions | 76.2 | 75.9 | 60.0 | 85.6 | 90.6 |
| low-level instructions | 85.6 | 85.7 | 66.7 | 88.0 | 91.0 |

The lowest accuracy is for long_press actions which is most likely due to the scarcity of these actions in the dataset. Detecting the name of the target app works well as well as learning the direction of a

scroll. In general, inferring action arguments, both target elements or input text, is much easier when the command is more explicit as in low-level instructions.

## E.2 Training with different levels of instructions

As shown in Table 11, fine-tuning with a mixture of HL and LL SeqIO tasks of ANDROIDCONTROL is equal or better than training individual models for different instruction levels. This is not a surprise as multi-task training increases the possibility of transfer learning. However, this may not always be true especially if different tasks contain conflicting data.

Table 11: Step-wise accuracy (%) on ANDROIDCONTROL of a LoRA-tuned PaLM 2S model using various configurations of instructions for training and testing. Each column represents a model each trained on a SeqIO task or a mixture of SeqIO tasks, while each row corresponds to an evaluation SeqIO task. HL and LL stand for high-level and low-level instructions, respectively.

|  | trained on HL | trained on LL | trained on HL+LL |
|---|---|---|---|
| tested on LL | - | 83.4 | **85.4** |
| tested on HL | **63.6** | - | **63.6** |

## E.3 Random-500 vs. full test split

Table 12 compares the difference of evaluating on Random-500 and the full test split. For zero-shot PaLM 2L, the step accuracy obtained on both splits are pretty similar, 35.2% vs. 35.0% for high-level instructions and 43.0% vs. 42.7% for low-level instructions. For fine-tuned PaLM 2S, the difference is larger but still smaller than the difference between a fine-tuned PaLM 2S and any zero-shot or few-shot model, so we consider it an accurate approximation for our analysis.

Table 12: Grounding accuracy on Random-500 and the full test split.

|  | Model | Test split | Step accuracy | |
|---|---|---|---|---|
|  |  |  | high-level | low-level |
| Zero-shot | PaLM 2L | Random-500 | 35.2 | 43.0 |
| Zero-shot | PaLM 2L | Full | 35.0 | 42.7 |
| Fine-tuned | PaLM 2S | Random-500 | 62.6 | 82.2 |
| Fine-tuned | PaLM 2S | Full | 64.8 | 80.0 |

## E.4 Step accuracy and episode accuracy vs. episode length

For this experiment we introduce the episode accuracy metric which measures the percentage of fully successful episodes. It is a harder metric since all step actions in a task must be predicted correctly for the task to be considered successful. We report this metric when testing on high-level instructions only, as for low-level instructions is less meaningful. Figure 9 depicts how both step accuracy and episode accuracy vary when increasing the episode length from 1 to 20 steps. We report performance of the PaLM-2S model fine-tuned on the full dataset (PaLM-2S-FT, rank=64) and the PaLM-2L zero shot model using the ER prompt (PaLM-2L-ZS) on the full test split. The episode length has no impact on the step accuracy because the difficulty of a single step is independent on the episode length. As a result, this is relatively flat (see solid lines in both graphs in Figure 9). However, as tasks become longer, the episode accuracy drops (dotted line in Figure 9b). For the zero-shot model, tasks longer than 5 steps are never completed. For the fine-tuned model the drop is more gradual, but when going from 5-step-task to 6-step-task the episode accuracy drops from 21.3% to 7.6% (despite the corresponding step accuracy being 64–71%).

## E.5 Step-accuracy performance vs. application types

We observe the screen representations provided as input to the UI control agent are critical to its success. Table 13 compares the step accuracy of various models on Google's first party apps and apps

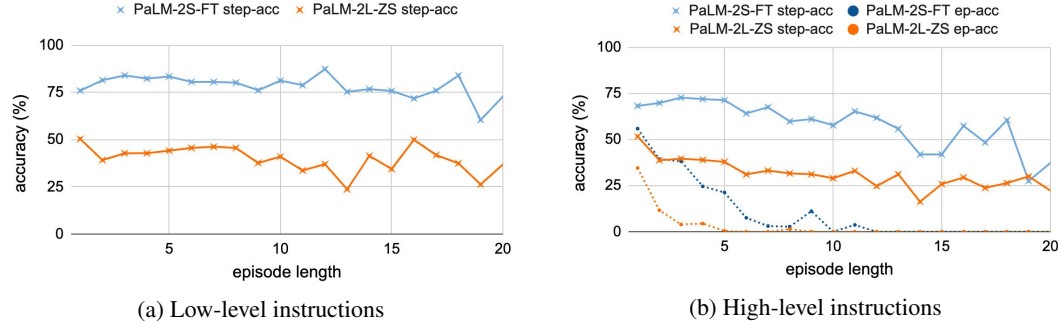

(a) Low-level instructions

(b) High-level instructions

Figure 9: Step accuracy and episode accuracy vs. episode length on the full test split. Tested models include PaLM-2S LoRA-tuned on the full dataset (PaLM-2S-FT) and PaLM-2L zero shot (PaLM-2L-ZS) using the ER prompt.

developed by third party developers (3rd-party). It is evident the gap in performance, especially in handling high-level instructions. All zero-shot methods perform significantly better on first-party apps than on third-party apps. This is evident for fine-tuned models, especially on high-level instructions where the most performance model (LT-all) achieves 82.5% step accuracy on first-party apps and only 58.7% accuracy on third-party apps. When the input is made of low-level instructions the gap is smaller and in some cases (few-shots) the performance on third-party apps is higher. This shows how for tasks that require stronger reasoning capabilities accurate screen representations are particularly critical.

Table 13: Step accuracy performance of some of the zero-shot, few-shot, and LoRA-tuned models we studied broken down by Google apps and non-Google (third party) apps. All tests are on Random-500. For all LoRA-tuned models rank=4 except for LT-all-r64.

| Regime | Method | Model | 1st-party | | 3rd-party | |
|--------|--------|-------|-----------|----|-----------|----|
| | | | hl | ll | hl | ll |
| Zero-shot | ER | PaLM 2S | 30.1 | 41.5 | 18.9 | 39.6 |
| | ER | PaLM 2L | 39.8 | 46.2 | 34.0 | 42.1 |
| | ER | GPT-4 | **45.6** | **54.7** | 28.7 | 46.6 |
| | ER | Gemini 1.5 | 39.8 | 52.8 | 24.4 | 42.9 |
| Few-shot | FS-5 | Gemini 1.5 Pro | **52.0** | 37.7 | 38.5 | 52.0 |
| | FS-10 | Gemini 1.5 Pro | 47.6 | 38.7 | 38.3 | 54.1 |
| | FS-100 | Gemini 1.5 Pro | 45.6 | 49.1 | 37.9 | **54.5** |
| LoRA-tuned | LT-5 | PaLM 2S | 38.8 | 57.5 | 26.2 | 56.3 |
| | LT-10 | PaLM 2S | 36.9 | 58.5 | 28.0 | 57.6 |
| | LT-100 | PaLM 2S | 48.5 | 59.4 | 35.8 | 61.9 |
| | LT-1k | PaLM 2S | 66.0 | 72.6 | 49.6 | 69.0 |
| | LT-10k | PaLM 2S | 74.8 | 79.2 | 54.7 | 77.9 |
| | LT-all-64 | PaLM 2S | **82.5** | **86.8** | 58.7 | 85.0 |

