# Supplementary Materials for
# On the Effects of Data Scale on
# Computer Control Agents

The paper and the Appendix of the paper provide details on the dataset (data format, statistics, usage, etc.). For completeness, in the following we include a datasheet based on the format of [1].

# 1 Datasheet

## 1.1 The ANDROIDCONTROL dataset

### 1.1.1 Motivation

- For what purpose was the dataset created? Was there a specific task in mind? Was there a specific gap that needed to be filled?
  - The dataset was created to support research in autonomous UI agents that interpret human natural language instructions and perform actions directly on a device UI. Each ANDROIDCONTROL task instance includes both high and low-level human-generated instructions, allowing researchers to explore the level of task complexity an agent can handle. Moreover, ANDROIDCONTROL is the most diverse computer control dataset to date, that facilitates in-depth analysis of the model performance in and out of the domain of the training data.
- Who created the dataset (e.g., which team, research group) and on behalf of which entity (e.g., company, institution, organization)?
  - The dataset was created by Wei Li, William Bishop, Alice Li, Chris Rawles, Folawiyo Campbell-Ajala, Divya Tyamagundlu, and Oriana Riva at Google.

### 1.1.2 Composition

- What do the instances that comprise the dataset represent (e.g., documents, photos, people, countries)? Are there multiple types of instances (e.g., movies, users, and ratings; people and interactions between them; nodes and edges)?
  - The dataset contains episodes of human demonstrations for mobile device control. It consists of natural language instructions at both the task level and step level, observations (screenshots + UI trees), and actions performed on Android devices. An example instance is shown in Figure 1.
- How many instances are there in total (of each type, if appropriate)?
  - The dataset comprises 15,283 episodes spanning 833 apps.
- Does the dataset contain all possible instances or is it a sample (not necessarily random) of instances from a larger set?
  - The dataset is a sample drawn from interactions across hundreds of Android apps. It does not cover all possible interactions, but it is representative of many use cases, as the instances were collected from diverse sources and varying conditions to emulate a real system's experience.

- What data does each instance consist of?
  - Each instance consists of a goal instruction in natural language, a sequence of observation-action pairs describing the execution of the task. Observations consist of full-resolution screenshots along with UI trees extracted from accessibility view hierarchies. Actions are described by: *type* (`click`, `long_press`, `input_text`, `scroll`, `navigate_back`, `navigate_home`, `open_app`, `wait`), *target UI element* (only for `click`, `input_text` and `long_press`), and action arguments (text for `input_text`, app_name for `open_app`, direction for `scroll`).
- Is there a label or target associated with each instance?
  - Yes, each episode has a natural language instruction associated.
- Is any information missing from individual instances?
  - No.
- Are there recommended data splits (e.g., training, development/validation, testing)?
  - Yes, there are recommended data splits of training, validation, and testing. The test split contains four sub-splits: *1) in domain data (IDD)*: randomly pulled episodes from the same distribution as the training data; *2) unseen-app*: a test split using apps not present in the train split; *3) unseen-task*: a test split with tasks not present in the train split; and *4) unseen-category*: a test split with apps from categories not present in the train split. Note that the test splits may contain overlapping episodes.
- Are there any errors, sources of noise, or redundancies in the dataset?
  - We have removed episodes that contained errors and annotators marked as failed. We are not aware of any errors in the released dataset.
- Is the dataset self-contained, or does it link to or otherwise rely on external resources (e.g., websites, tweets, other datasets)?
  - It is self-contained.
- Does the dataset contain data that might be considered confidential?
  - No.
- Does the dataset contain data that, if viewed directly, might be offensive, insulting, threatening, or might otherwise cause anxiety?
  - No, the dataset does not contain anything offensive, insulting, threatening, or that might cause anxiety.

### 1.1.3  Collection process

- How was the data associated with each instance acquired?
  - The data associated with each instance was acquired by using a physical Android phone (Google Pixel with Android 8.0 or higher) installed with a companion Android app that in turn connects to a web app running on a desktop Chrome browser. Annotators control the phone through the web app, using the WebUSB protocol and Android Debug Bridge (ADB). The web app provides annotators with controls to perform actions on the phone and observe their outcome. An annotator select UI actions to perform on the phone. Before executing each action, the annotator is asked to type in a short natural language description of the action they are about to take.
- What mechanisms or procedures were used to collect the data (e.g., hardware apparatus or sensor, manual human curation, software program, software API)?
  - The mechanisms used to collect the data include a hardware apparatus (Android device), manual human curation (annotators collected interaction traces), and a software program (for executing the tasks and capturing data).
- Over what timeframe was the data collected?
  - The data was collected from April 2023 through April 2024.
- Does the dataset relate to people?
  - The dataset does not directly relate to people, however annotators collected the data. No personal information was included in the task specification or demonstration collection.

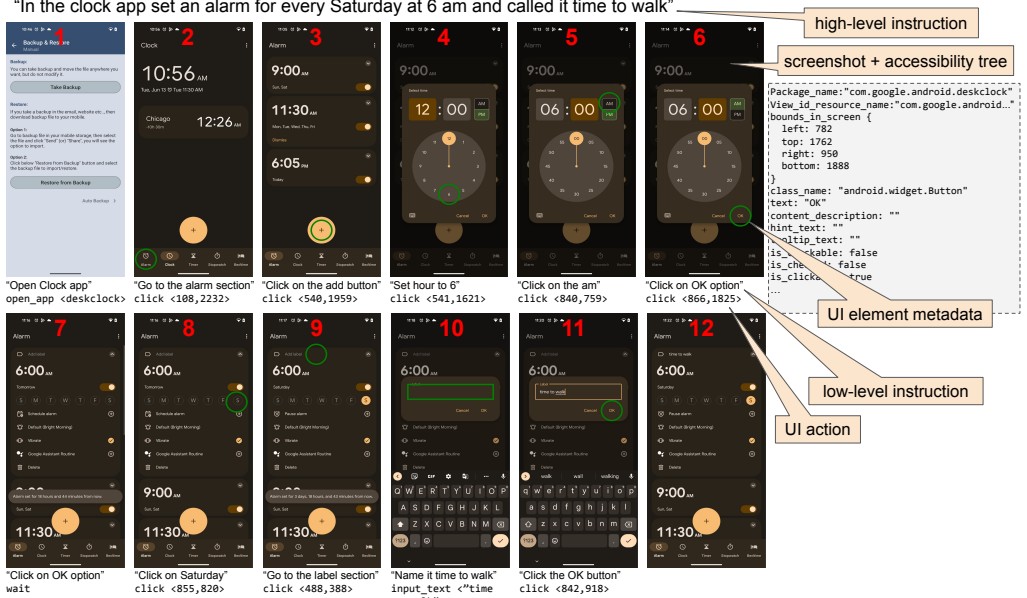

Figure 1: An example task demonstration contained in ANDROIDCONTROL. Green circles/rectangles highlight the action on the screen. Red numbers are added only for illustration purposes.

### 1.1.4 Preprocessing

- Was any preprocessing/cleaning/labeling of the data done (e.g., discretization or bucketing, tokenization, part-of-speech tagging, SIFT feature extraction, removal of instances, processing of missing values)?

  – Preprocessing of the data was done. Scroll actions were processed to be reduced to four scrolling directions (left, right, up, and down) using information of start position, end position, and speed.

- Was the "raw" data saved in addition to the preprocessed/cleaned/labeled data (e.g., to support unanticipated future uses)?

  – Raw, full-resolution screenshots are provided. We do not provide raw logged events and discarded episodes.

- Is the software used to preprocess/clean/label the instances available?

  – The data preprocessing and cleaning was done using Python. We do not release this part of the code.

### 1.1.5 Use cases

- Has the dataset been used for any tasks already?

  – Yes, we have trained and evaluated several models on this dataset for device automation. These models can serve as baselines for future research.

- Is there a repository that links to any or all papers or systems that use the dataset?

  – Yes, at `https://github.com/google-research/google-research/tree/master/android_control`.

- What (other) tasks could the dataset be used for?

  – It is currently used for device automation. Future use cases could include screen understanding, screen generation, question answering, image captioning, and activity recognition.

- Is there anything about the composition of the dataset or the way it was collected and preprocessed/cleaned/labeled that might impact future uses?

- The set of app categories in ANDROIDCONTROL is still an incomplete representation of all tasks users may ask agents to perform.

### 1.1.6 Distribution

- Will the dataset be distributed to third parties outside of the entity (e.g., company, institution, organization) on behalf of which the dataset was created?
  - The dataset is free to all under the condition that the dataset is used for non-commercial purposes only.
- How will the dataset be distributed (e.g., tarball on website, API, GitHub)?
  - The dataset is available at `https://github.com/google-research/google-research/tree/master/android_control`.
- Will the dataset be distributed under a copyright or other intellectual property (IP) license, and/or under applicable terms of use (ToU)?
  - The dataset is released under the Apache License, Version 2.0.[1]
- Have any third parties imposed IP-based or other restrictions on the data associated with the instances?
  - N/A.

### 1.1.7 Maintenance

- Who is supporting/hosting/maintaining the dataset?
  - The dataset is maintained by the authors.
- How can the owner/curator/manager of the dataset be contacted?
  - The contact information of the authors can be found at the beginning of the main paper and will be provided in the data repository.

### 1.1.8 More examples

Examples of episodes from ANDROIDCONTROL are shown in Figure 2.

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

Share my favorite Book "the Queen's Gambit" to my Friend Natalie larson over her gmail address -natalie.larson1998@gmail.com from the PocketBook app.

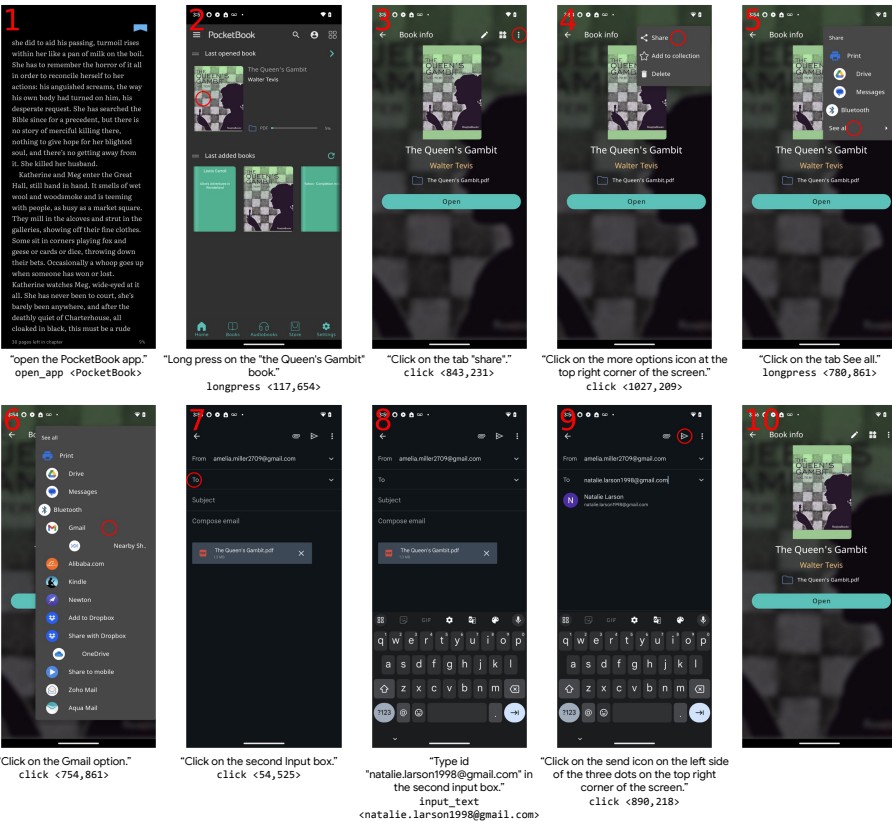

Launch the Amazon app, search for India gate basmati rice, and add 5 Kg of it to your shopping cart.

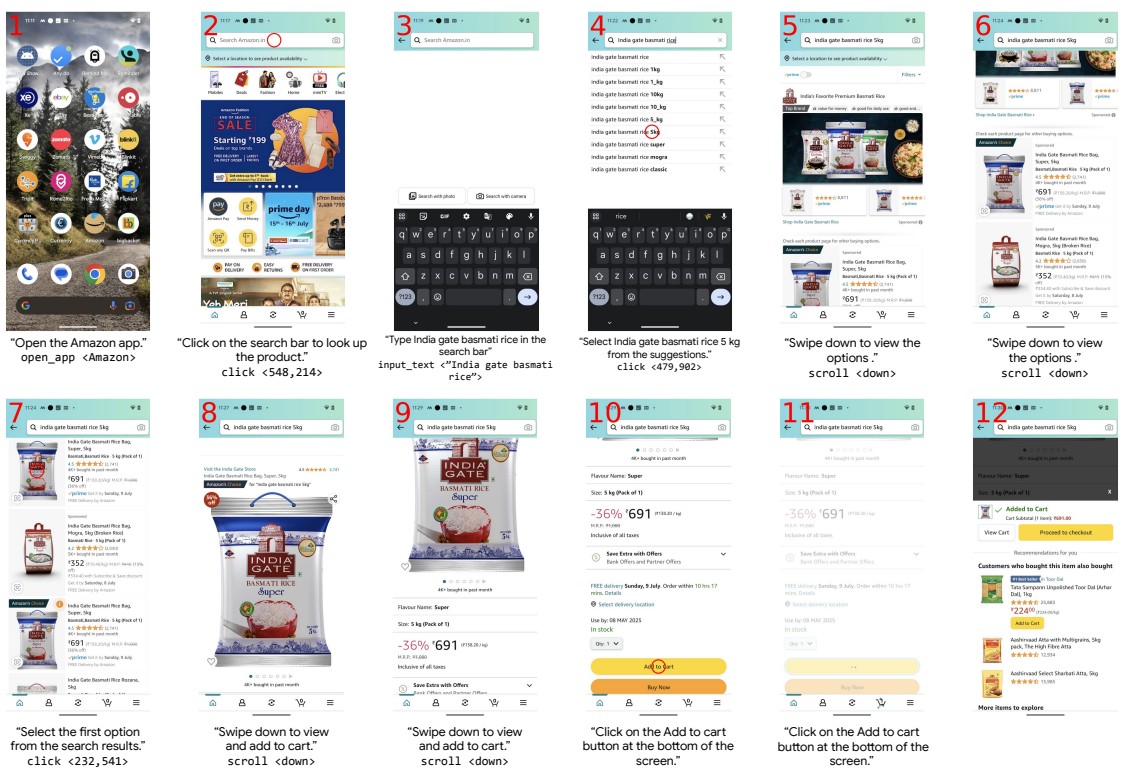

Figure 2: Example episodes contained in ANDROIDCONTROL. Red circles highlight the location of click and long press actions on the screen. Red numbers are added only for illustration purposes.