# OpenReview forum: "On the Effects of Data Scale on UI Control Agents"
_NeurIPS.cc/2024/Datasets_and_Benchmarks_Track — NeurIPS 2024 Track Datasets and Benchmarks Spotlight_

### Official Review · Reviewer_CboS · 2024-07-22

**Rating:** 7
**Confidence:** 5
**Correctness:** Yes.
**Clarity:** Yes.

**Review:**

Overall this is a quite good paper, which considers the effect of data scale on fine-tuning mobile GUI agents for the first time, which is quite interesting and insightful. The writing, illustration, and experiment design of the paper is also very good. The dataset also includes both high-level and low-level human-generated instructions for each task.



Pros:

- The paper considers the effect of data scale on fine-tuning mobile GUI agents for the first time, which is quite interesting and insightful.
-  The dataset includes both high-level and low-level human-generated instructions for each task.
- The writing and illustration of the paper is good.
  - The comparsion with previous works is clear.
  - The illustration of the statistics of the benchmark.
- The experiment design and the systematical evaluation is solid.
- The conclusion from the experiment is quite insightful.

Cons:

- As a paper on dataset & benchmark track, although the research question of "the effect of data scale on fine-tuning mobile GUI agents" is very meaningful, the dataset AndroidControl proposed in the paper is not good enough. Compared with previous works, although the data quality is higher than AITW, the data quantity is much smaller. In addition, the average task step is only 4.8, which indicates that the tasks in the dataset are much simpler than previous works.
- The term "Computer Control Agent" in the title and text of the paper is quite confusing. I think it should be "Mobile GUI Agent", "Mobile Device Control Agent", etc., instead of the term "Computer".

**Strengths:**

The paper considers the effect of data scale on fine-tuning mobile GUI agents for the first time, which is quite interesting and insightful; The dataset includes both high-level and low-level human-generated instructions for each task; The writing and illustration of the paper is good; The experiment design and the systematical evaluation is solid; The conclusion from the experiment is quite insightful.

**Additional Feedback:**

N/A

**Documentation:**

Yes.

**Ethics:**

No.

**Limitations:**

Yes.

**Opportunities For Improvement:**

- The dataset AndroidControl proposed in the paper is not good enough. Compared with previous works, although the data quality is higher than AITW, the data quantity is much smaller. In addition, the average task step is only 4.8, which indicates that the tasks in the dataset are much simpler than previous works.

- The term "Computer Control Agent" in the title and text of the paper is quite confusing.

**Relation To Prior Work:**

Yes.

**Summary And Contributions:**

This paper presents AndroidControl, a mobile GUI agent dataset containing 15,283 unique tasks on 833 Android applications. The dataset includes both high-level and low-level human-generated instructions for each task. The authors then use this dataset to study the effect of data scale on fine-tuning mobile GUI agents. PaLM-2S is used to perform fine-tuning. Experiment results show that the fine-tuned model outperforms zero-shot and few-shot baselines when tested in-domain, and that robust performance can be achieved by simply collecting more data. Out-of-domain performance scales significantly slower.

---

> ### Author Rebuttal · Authors · 2024-08-16
>
> We thank the reviewer for their insights and appreciation of AndroidControl.
>
> Before providing a response to the reviewer’s questions, we would like to report a small correction to the stats reported in Table 1 in our paper. The number of unique instructions in our dataset is not 15,283 but instead 14,548. This correction comes about from incorrectly reporting the number of episodes (which was correctly reported as 15,283) also as the number of unique instructions in Table 1.  We will update the manuscript, and we apologize for any confusion it may have caused. We thank Reviewer SNrC for pointing out that the two numbers were identical thus aiding us in catching this mistake.
>
> **AndroidControl relative to other datasets**
>
> We refer the reviewer to our response regarding how AndroidControl complements existing datasets in the global response.
>
> **Confusing title**
>
> The reviewer has a good point. We debated on how to name this work and decided to use “computer control agent” simply because it is a relatively established term. However, we agree with the reviewer that it is confusing and not specific to our work. We will replace it with a new term, possibly “mobile control agent”. Thank you for pointing this out.

---

> > ### Comment · Reviewer_CboS · 2024-08-28
> >
> > Thanks for your effort. I will raise my original rating.

---

### Official Review · Reviewer_Lq7f · 2024-07-24
**Useful dataset and insightful findings**

**Rating:** 8
**Confidence:** 4
**Clarity:** This paper is well written and easy t…

**Review:**

Overall I think this is a solid work and I would like to see it being published.

**Strengths:**

The dataset exhibits both high quality and good diversity, and can be very useful for the research community in developing on-device agents. It supports the evaluation on in-domain data, unseen-app, unseen-task and unseen-category, which sheds light on the different scaling patterns of LLMs' fine-tuning performance on each setting.

**Additional Feedback:**

For LORA-tuned experiments, are the episodes randomly sampled, or some data selection methods are applied? If so, what are the selection criteria?

**Correctness:**

The dataset is constructed in a sound way, and the experiment design are reasonable to me.

**Documentation:**

Yes.

**Ethics:**

I don't have any ethical concerns about this work.

**Limitations:**

The limitations are discussion in the last section.

**Opportunities For Improvement:**

I can see the great potential for the dataset to be used for developing on-device agents that helps automate frequently used workflows. It would be helpful to discuss how the fine-tuning performance/inference cost scale with model size and quantization bits too, as on-device models are subject to limited computational resources.

**Relation To Prior Work:**

The prior work is clearly covered and the proposed dataset is well compared with the prior work.

**Summary And Contributions:**

This paper introduces AgentControl, a dataset consisting of 15,283 demonstrations of everyday tasks over 833 Android apps. The dataset is collected by trained annotators over the course of a year. The Android apps it covers are from 40 different categories, including Google apps (Settings, Gmail, Google Maps, etc.), high-trend apps (e.g., Amazon, Booking.com, Kayak, Spotify, etc.) as well as less-popular or regional apps. The authors conducted a study on how fine-tuning on AgentControl scales in different settings, including high-level task vs. low-level task, and in-domain vs out-of-domain. They also compared fine-tuning with zero-shot and few-shot baselines, with a finding that fine-tuning scales well in domain, but less-favorably out of domain.

---

> ### Author Rebuttal · Authors · 2024-08-16
>
> We thank the reviewer for their helpful response and strong appreciation of our work.
>
> Before providing a response to the reviewer’s questions, we would like to report a small correction to the stats reported in Table 1 in our paper. The number of unique instructions in our dataset is not 15,283 but instead 14,548. This correction comes about from incorrectly reporting the number of episodes (which was correctly reported as 15,283) also as the number of unique instructions in Table 1.  We will update the manuscript, and we apologize for any confusion it may have caused. We thank Reviewer SNrC for pointing out that the two numbers were identical thus aiding us in catching this mistake.
>
> **Discussion of fine-tuning performance/inference cost**
>
> Thank you for your suggestion. You’re absolutely right. Especially for an Android control agent, the ability to run on-device is of paramount importance (more than for web or desktop control agents). The base model we fine tuned in this experiment (PaLM2S) is relatively small, but we are also trying other smaller models like Gemini Nano and PaliGemma and are also interested in strategies where one LLM call predicts multiple next steps, which would reduce inference cost.  We will certainly include a discussion on the on-device strategy we are envisioning and the trade-offs to consider in the camera ready version.
>
> **Subsampling method for LoRA-tuned experiments**
>
> For LORA-tuned experiments, when we trained on a subset of the training data, episodes were selected at random. Thank you for pointing out this was not clear; we will clarify this in the camera ready version.

---

### Official Review · Reviewer_SNrC · 2024-07-24
**Review of AndroidControl Dataset**

**Rating:** 7
**Confidence:** 3
**Correctness:** To the best of my knowledge, the clai…

**Review:**

I like how the authors did a solid job in conducting as many experiments as feasible, and I was able to get some insight from the paper. The dataset itself would also be useful for the community. However, at the same time I also have some concerns, some on the dataset itself, and some on the details of the experiments. See my Limitations section.

Overall, I am still impressed by the paper and am giving an acceptance rating of 6. If the authors are able to address my issues, I'd be happy to raise my score.

**Strengths:**

- AndroidControl has a diverse number of apps and the second most number of demonstrations when compared to baseline datasets.
- The authors conduct extensive experiments to investigate the effect of dataset size and training methodologies on model performance. The results act as solid evidence to backup the authors' hypotheses.

**Additional Feedback:**

All my comments, suggestions, and questions can be found in the other sections.

**Clarity:**

In general, the paper is clearly written and I found it easy to follow and understand. However, there are quite a few typos or mistakes:
- Figure 1's high level instruction is grammatically incorrect. The end of the sentence should be "call it time to walk" instead. Also, the bubble for UI element metadata is confusingly pointing towards the OK button in image 6, instead of the metadata box.
- Line 144: "first-party apps such Clock or Settings" missing an "as"
- Line 243: "Rawles et a.. [26]" should be *et al.*
- Table 4: "Performance on the IID sub-split" should be "on the IDD sub-split". The use of the IDD acronym is actually somewhat confusing. Not only does it remind me of the i.i.d. acronym, but also the fact that the authors use "in-domain" in other places, e.g., Figure 5a's title, makes references to the concept inconsistent in the paper.

**Documentation:**

The authors provided a [URL](https://github.com/google-research/google-research/tree/master/android_control) to a Github page, which contains an Apache license, as well as links to the actual dataset files. Currently, the only documentation is the README file, but the file gives installation directions, code examples, as well as the data structure descriptions.

**Ethics:**

I do not see any ethical concerns in this work.

**Limitations:**

The authors have addressed some of their own limitations.
- Only one model was fine-tuned.
- Evaluation was offline, i.e. does not reward alternative routes or correction.

However I still have the following concerns/questions about the experiments in the paper:
- While AndroidControl excels the other datasets in at least one metric, I don't exactly see why everyone should use AndroidControl over existing datasets, i.e. it is not fundamentally introducing something new in the dataset. Even quantity-wise there are datasets that
- In Table 3, the average number of elements per screen experience a sudden decrease on the app-unseen, task-unseen, and category-unseen test sub-splits. Can the authors explain this phenomenon? The elements on the screen shouldn't be that different from the train or validation set.
- Looking closely at Figure 5, the in-domain trend and out-of-domain trend seem identical until 10k episodes. While this is the message which the authors are also trying to convey, it also makes me feel that linear extrapolation to estimate performance implies too many assumptions. For example, how do we know that this trend (of in-domain performance higher than out-of-domain) will continue after 10k? Only the data points around 10k show a difference (seems more like extrapolating on a single point than on a trend).
- On line 221-222, the authors mention rank switched from 4 to 64 at 10k. This does not seem to align with results in Figure 5, where rank becomes 64 starting at 1k.

**Opportunities For Improvement:**

- The number of human demonstrations and unique instructions are exactly the same, as given in Table 1. If I'm understanding it correctly, this means there is only one demonstration for each task. I don't know how "uniquely defined" a task is, but this makes me worry about the diversity in each task, and if trained on a specific task whether or not there will be enough data.
- The authors note in line 198 that the agent implementation does not directly leverage the page screenshot. While they argue that UI tress are a better representation, this means that the quality of the screenshots in the dataset are not evaluated in any way.

**Relation To Prior Work:**

Table 1 lists out 10 prior datasets for computer control. AndroidControl distinguishes itself by supporting both UI tree and screenshot for the UI state, as well as supporting both high-level and low-level instructions for tasks. Compared to those which also possess these features, AndroidControl offers a greater number of demonstrations, unique instructions, as well as apps.

**Summary And Contributions:**

The AndroidControl dataset has 15283 unique tasks over 833 Android apps and aims to be used as a benchmark for Computer Control Agents. The contributions of AndroidControl include
- The dataset itself with high-level and low level instructions, diverse number of unique tasks, and diverse number of apps.
- The authors measured the effects of scaling during fine-tuning, both in and out of domain, as well as for high and low-level instructions.
- The authors compare the performance between fine-tuning vs. zero-shot or few-shot prompt tuning. The results show that while fine-tuning is the best method in domain, it does not exceed prompt tuning that much in performance on lower scale datasets for high-level instructions.

---

> ### Author Rebuttal · Authors · 2024-08-16
>
> Thanks for the helpful feedback. We respond to each point below.
>
> **Number of unique instructions**
>
> First, we would like to report a small correction to Table 1. The number of unique instructions in our dataset is 14,548 (not 15,283). We will update the manuscript, and we apologize for any confusion.
>
> The reason the number of unique AndroidControl task instructions and episodes are similar is that task descriptions are not derived from a template but instead generated by human annotators. We manually define general task categories and ask annotators to provide multiple instantiations of them. For example, for the task category “watch videos”, some of the tasks they created are:
>
> - In YouTube open the video change your brain by Andrew Huberman and play it
> - Open YouTube. Open the video of lex fridman and play it.
> - In one of the event, i tried mediterranean flatbread and i liked the taste of it so i want to watch its food recipe in vimeo.
> - Play the song "I Don't Wanna Go Out" by "Wet Leg" on the YouTube app.
> - Play the video "Dina: A Passion for Italian Food" for Me on the Vimeo app.
>
> For a more advanced task category like “Watch a video. You can pause, rewind, and fast-forward the video. You can also add videos to your Watch Later playlist.” we obtained tasks as the following ones:
>
> - Go to vimeo app and play the video atif aslam mashup then move forward the video by 20 seconds and pause the video. Add the video to the watch later playlist.
> - Play the video Roar by katy perry then move forward the video by 10 seconds and pause the video. Add the video to the watch later playlist
>
> As shown above, the high-level descriptions are unique and the complexity can vary quite a lot. There is enough data for the model to learn every task category since even if expressed differently, the actual executions have similarity. For low-level task descriptions the overlap is of course higher. The low-level instructions for the first two high-level instructions above are the following:
>
> - swipe down to minimize the video, close the video, clear the search bar, search for change your brain by Andrew Huberman, click on the search icon, play the first video
> - go back, go to the search bar, clear search bar, search for lex fridman, click on search icon, play the first video, skip the ads
>
> **Screenshot quality**
>
> We carefully recorded and saved the screenshots at high resolution (2400 x 1080), which can be verified by visual examination of the examples in the main paper and Appendix.
>
> **Why AndroidControl?**
>
> Please see the general response.
>
> **Number of UI elements in test splits**
>
> We note that in the test sub-split that is sampled from the same distribution as the train split (the “IDD” split as it is currently worded in the paper), the number of UI elements per screen are comparable, as expected. To explain why the other sub-splits demonstrate a drop in the number of UI elements per screen, we calculated the average number of UI elements across all screens in the data for each app (for simplicity we call this “the average number of UI elements per app”).  Fig. 1 (in the attached PDF) shows the distribution of the average number of UI elements per app for the entire dataset. The distribution shows a long tail, with a few apps with a very large number of UI elements per screen. If we remove these outliers by dropping the top 5% of apps with the most elements per screen, the average number of UI elements across the remaining apps is 180.9, which is inline with the unseen test sub-splits. A reasonable question is then why do the unseen test sub-splits not sample from the tail? We believe this is because, as shown in Table 3 of the paper, the number of apps in each sub-split is relatively small. Further, as stated in line 182 of the paper, there is overlap between the different test sub-splits (this was done to keep the total size of the test set reasonable, if each sub-split needed to be disjoint, it would leave less train data), so the total number of apps across the union of all unseen test-subsplits is still modest, again supporting that the we simply didn’t draw enough outliner apps from the long tail to affect the average for the unseen splits. We will clarify these points in the paper and include Fig. 1.
>
> **Fig. 5: in-domain and out-of-domain trends**
>
> To better compare the in-domain and out-of-domain trends we plotted them in Fig. 2 (see attached PDF). For low-level instructions, the in-domain and out-of-domain data points are indeed close and then diverge. The similarity of the trends for the low-level instructions is not surprising, as different high-level tasks (even those from different domains) will often reuse the same types of low-level actions, making generalization for low-level commands an easier problem.  For high-level instructions, the in-domain trendline grows faster while the out-of-domain trendline has a smaller slope.  While it is impossible to say if the linear trend would continue beyond the amount of data we were able to test with, we did measure the goodness of our approximations by computing R2 coefficients (reported in the figure). All coefficients are above 0.95, suggestions that they are not influenced heavily by a single outlier.  We will include these R2 values in the paper.
>
> **LoRA rank**
>
> The reviewer is correct. The caption of Figure 5 is correct, but the text at line 221-222 should be “< 1k episodes” instead of “< 10k episodes”. Thank you!
>
> **Fig. 1**
>
> The high-level instruction in the figure is a real instruction from the dataset. We left the typo because annotators often introduce errors of this type. We’ll clarify this in the caption. The “UI Element metadata” bubble points to the “OK button” because the metadata shown in the gray box is for that UI element. We’ll improve the visualization. Thanks for letting us know this is confusing.
>
> **IDD acronym + typos**
>
> Thank you for your careful review - we will correct all typos and clarify our usage of IDD and ODD.

---

> > ### Comment · Reviewer_SNrC · 2024-08-28
> >
> > Thanks for your effort, a lot of my concerns were addressed. I have raised my score to 7.

---

### Author Rebuttal · Authors · 2024-08-16

We thank the reviewers for their time and insightful comments. We have responded individually to each reviewer, but as both Reviewer SNrC and Reviewer CboS had questions related to the quality and size of AndroidControl we respond to this here. Reviewer SNrC noted that AndroidControl excels the other datasets in at least one metric, but asked why everyone should use AndroidControl. Reviewer CboS noted that the data quality of AndroidControl is higher than that of AitW, but also that the data quantity is much smaller and questioned the task complexity.

First of all, we would like to clarify that we do not propose that AndroidControl should be used in exclusion of existing datasets. Instead, given the high cost (hiring humans to perform tasks) of collecting such datasets we feel that the community will be best served by having multiple datasets, with some exceeding in some metrics and others in other metrics (since it is unlikely, for cost reasons, any one dataset will sufficiently cover all aspects we would want to measure UI control agents against).

It is in this spirit of complementing existing datasets that we are releasing AndroidControl. The particular strengths of AndroidControl are:

1. Data diversity: compared to all the existing datasets AndroidControl is the most diverse in terms of unique instructions and unique apps.
2. Data completeness: unlike AitW that contains only app screenshots, AndroidControl also contains Android accessibility trees. Moreover, for every high-level instruction, it contains the sequence of low-level instructions generated by human annotators (not synthetically).
3. Size: If we exclude MiniWob++ which consists of synthetic (not real) websites and UIBert which includes only single-step instructions, AitW is the only dataset that is larger than AndroidControl. In addition, we believe there are substantial strengths of AndroidControl that AitW lacks, which we detail next.

The authors of this paper are also the authors of the AitW paper. After collecting AitW and using it for 2+ years in the context of a behavioral cloning project, we ourselves ran into several limitations of AitW.  We started the collection of AndroidControl inspired by these limitations. AitW does not contain UI trees which affected how well our models could learn to map natural language commands to UI elements. The OCR and IconNet information it provides are also very noisy (Figure 2 in https://arxiv.org/pdf/2407.17490, authored by others, shows a clear example of that). The inclusion of accessibility trees in AndroidControl allowed us to extract more precise bounding boxes, thus improving model learning for grounding. Moreover, there is extremely high redundancy of task instructions in AitW, which has led others to pair down the dataset to a subset with diverse tasks, but this yielded only 2.5k episodes (“Android in the Zoo (AitZ)”, https://arxiv.org/abs/2403.02713).

In addition, AitW achieves much of its diversity through the addition of tasks in the Chrome browser.  There is no problem, per se, with this, but it does mean that the number of Android apps in the dataset are limited (despite its size in total number of episodes).  This is a primary place where we feel AndroidControl complements AitW, as automation of non-Chrome based apps is a very important problem.  To illustrate this point, in the attached PDF we provide a table  comparing key statistics of AitW and AndroidControl, when Chrome-based tasks are removed from each one.

While AitW has a much larger number of episodes, it is only for one fifth of the apps present in AndroidControl and for a much smaller set of unique tasks (1.6k vs. 14.5k). In other words, while AitW is a very large dataset, many of its episodes are for the same identical tasks, hence its coverage of Android tasks and apps is very limited. AndroidControl’s high-level instructions are also much longer (19.1 vs. 7.8 words) which is an indication of its tasks being generally more complex.

Regarding the length of tasks in AndroidControl, Table1 in the paper shows that other datasets contain longer tasks. The length of tasks can depend on many factors and it is not always an indication of the task complexity.

The length of a task can depend on the scenario. WebLINX (https://mcgill-nlp.github.io/weblinx/) tasks have 43 steps because they simulate multi-turn conversations, therefore chaining together many sub-tasks. WebShop contains shopping tasks of 11.3 steps which consist of sequences of searches and filter selections; while the tasks are longer, the variety of actions is very limited as it is constrained to one app and even one task type. AndroidControl focuses on single-turn scenarios for a large variety of task categories.

The length of tasks in a dataset can also depend on the collection methodology and specifically whether annotators were allowed to recover from errors while recording a task. AndroidControl contains only successfully-completed tasks; when annotators make errors we ask them to mark the trace as failed. Failed traces are excluded from the release and from training. If annotators were allowed to recover from errors, tasks would be longer but possibly noisier.

Finally, the length of tasks can depend on the data collection infrastructure. For example, AitW tasks are slightly longer than those of AndroidControl (6.5 vs. 4.8 steps) but we attribute this to the following two factors instead of task complexity:
- The annotators of AitW operate directly on Android emulators, and the actions are recorded through device instrumentation. In other words, any unnecessary touches are also recorded as steps. In contrast, AndroidControl requires clicking buttons on a separate GUI to explicitly trigger any recording of actions, hence discourages unnecessary actions.
- Each contiguous scroll gesture of AndroidControl is combined and quantized into a single action, while for AitW it is recorded as a sequence of dual-point drags.

---

### Decision · Program_Chairs · 2024-09-26

**Decision:**

Accept (Spotlight)

**Comment:**

This paper presents a novel dataset AndroidControl, consisting of approximately 15,000 unique instructions and demonstrations for tasks on the Android platform. The set of tasks span over 800 apps, and record both the screenshot, the accessibility tree, and natural language instructions (both high and low level for the task). The authors also present an investigation in to how the finetuning of models changes based on scale of data, looking at both in domain and out of domain performance.

*Strengths:*

* The paper was commended for its many rigorous experiments and its solid empirical study.
* The paper was also commended for the diversity of the dataset (including number of apps/tasks) and its scale.
* The writing, and illustration of the paper was also commended. Multiple authors said they would like to see this published/could see the paper being very useful to the community.
* this paper addresses one of the crucial questions for agentic systems.

*Weaknesses:*

* Some questions were raised about the size & quality of Android Control, in particular in comparison to other datasets like AitW.
The authors provided a substantial rebuttal by highlighting the diversity and new modalities of this dataset (by including accessibility tree), and adressing the advantages of AC in comparison to AitW.

*AC Note:*

I agree with the unanimous decision of the reviewers that this novel dataset and empirical study will be of great use to the community seeking computer control agents, and recommend acceptance.